# Hardware-aware training for large-scale and diverse deep learning inference workloads using in-memory computing-based accelerators

Malte J. Rasch ●[1] ✉, Charles Mackin ●[2], Manuel Le Gallo ●[3], An Chen ●[2], Andrea Fasoli ●[2], Frédéric Odermatt ●[3], Ning Li ●[1], S. R. Nandakumar[3], Pritish Narayanan[2], Hsinyu Tsai ●[2], Geoffrey W. Burr ●[2], Abu Sebastian ●[3] & Vijay Narayanan ●[1]

Analog in-memory computing—a promising approach for energy-efficient acceleration of deep learning workloads—computes matrix-vector multiplications but only approximately, due to nonidealities that often are non-deterministic or nonlinear. This can adversely impact the achievable inference accuracy. Here, we develop an hardware-aware retraining approach to systematically examine the accuracy of analog in-memory computing across multiple network topologies, and investigate sensitivity and robustness to a broad set of nonidealities. By introducing a realistic crossbar model, we improve significantly on earlier retraining approaches. We show that many larger-scale deep neural networks—including convnets, recurrent networks, and transformers—can in fact be successfully retrained to show iso-accuracy with the floating point implementation. Our results further suggest that nonidealities that add noise to the inputs or outputs, not the weights, have the largest impact on accuracy, and that recurrent networks are particularly robust to all nonidealities.

The ever-increasing compute needed to train and use deep neural networks (DNNs)[1] have made hardware latency and energy efficiency a growing concern. However, conventional processor architectures (e.g., CPUs, GPUs, etc.) incessantly transfer data between memory and processing through the "von Neumann bottleneck", inducing time and energy overheads that significantly degrade latency and energy efficiency. Numerous hardware concepts have been introduced to accelerate DNN training and/or inference[2–4], by approximating matrix-vector multiplications (MVMs) and other arithmetic with custom floating-point representations such as bfloat16[5] or DLFloat[6], or with reduced-precision fixed-point arithmetic to quantize synaptic weights and activations[7–10]. Model compression and sparsification techniques can further reduce compute requirements by pruning weights and/or activations[11,12].

Analog in-memory computing (AIMC) using non-volatile memory (NVM) elements is a promising mixed-signal approach for DNN acceleration[13–15], with weights stored using crossbar arrays of tuneable conductive elements. This enables approximate MVM computation directly in-memory, by applying activation vectors (as voltages or pulse durations) to the crossbar array, and then reading out analog physical quantities (instantaneous current or accumulated charge)[16–18]. As a "non-von Neumann" architecture, AIMC performs MVM operations at the location of the stored weights, in a highly parallel, fast, and energy-efficient manner[17]—but only approximately.

[1]IBM Research, TJ Watson Research Center, Yorktown Heights, NY, USA. [2]IBM Research Almaden, 650 Harry Road, San Jose, CA, USA. [3]IBM Research Europe, 8803 Rüschlikon, Switzerland. ✉e-mail: malte.rasch@ibm.com

The success of reduced-precision digital accelerators proved that DNNs can tolerate surprisingly coarse approximations of the underlying MVMs. While naive direct weight-quantization invariably leads to DNN accuracy loss, original model accuracies can often be recovered when DNNs are retrained in a quantization-aware manner, even for aggressive reductions in precision. Weight-quantization into as few as 2−4 bits can often be tolerated without significant accuracy reduction[8,19,20]. This observation led to the development of quantization-aware training (QAT) methods, now commonly used when deploying DNNs onto reduced-precision digital hardware[21].

In general, since reducing MVM precision decreases the representational power of each DNN layer as compared to a full floating-point (FP) implementation, accuracy naturally suffers once the function approximation becomes too coarse for the task at hand[20]. In practice, QAT is known to have limits, and MVM minimum-precision requirements vary across each DNN topology. For instance, the first and last layers of convolutional neural networks (CNNs) are invariably implemented with high precision FP, even in studies claiming CNN iso-accuracy at very low fixed-point precision[7,8].

Up to now, it has been unclear how and to what degree DNNs can be retrained to maintain accuracy on emerging AIMC technology. The successes of QAT cannot be directly translated onto AIMC, since the MVM approximations arise from fundamentally different concepts. In AIMC, weights are represented by conductances that are physical properties of NVM devices. In many materials, such as phase-change memory (PCM)[22,23], resistive random-access memory (ReRAM)[24,25], conductive bridge RAM (CBRAM)[26,27], or electro-chemical random-access memory (ECRAM)[28,29], these conductances are effectively continuous physical quantities, and stored weights are not quantized.

That said, effective AIMC weight precision is impacted by various nonidealities, including thermal and 1/f noise, randomization during physical switching induced by electrical and thermal fluctuations, material inhomogenities[30], and device-to-device variability introduced during device fabrication or operation. These issues cause both MVM read-out[31] and the writing or programming of the conductances[32–34] to be erroneous and non-deterministic. Worse yet, conductances can evolve over time after programming[35–37]. Finally, any nonlinearities within the analog circuitry performing summation can further degrade MVM precision. Such imperfections include "IR-drop" voltages on wires and transistors, restrictions on input (output) dynamic range imposed by discretization and saturation of the digital-to-analog converter (DAC) (analog-to-digital converter (ADC)) components, and random noise or variability in the circuitry.

Whereas QAT gets challenging as precision is deterministically reduced, MVM approximation in AIMC is tied to non-deterministic signal-to-noise ratio. A number of previous studies have shown that noise-aware training −simple injection of noise onto weights or activations during DNN training−can make DNNs more robust for AIMC deployment[33,38–42]. However, such studies have typically been limited to one or two exemplary DNNs of a particular type (e.g., CNN) using only a limited subset of nonidealities such as NVM noise. Other critical AIMC system aspects such as output noise, saturation, and circuit nonlinearities have been neglected. Moreover, since each study makes different hardware and NVM-device choices, it is difficult to generalize, compare, or combine them. Thus more realistic and standardized AIMC crossbar models−which can support comparison of AIMC accuracy for hardware-aware trained DNN models across studies−are needed.

Although some promising, small-sized DNN prototype demonstrations exist[43–49], it remains unclear how robust the AIMC deployment of realistically sized AI workloads will be. How will the various nonidealities of AIMC hardware impact the DNN accuracy, across all the various topologies and thus application domains? And how much of the lost accuracy could be recovered by hardware-aware training? Which crossbar-array design choices will be most effective in maintaining accuracy? And if necessary, what degree of improved device-to-device uniformity might be required−through better NVM-device fabrication−in order for AIMC to succeed on all DNN models? A systematic study comparing the various DNN topologies in terms of robustness to AIMC nonidealities is needed.

In this paper, we establish a robust hardware-aware (HWA) framework by extending and improving existing training methods for AIMC to include previously neglected nonidealities (see Fig. 1 for an illustration). We define a standard inference model for PCM-based AIMC that can readily be extended to other types of NVM devices. We explore the functional suitability of AIMC across application domains by assessing the robustness of a wide set of DNN topologies. Finally, we estimate the individual impact of various AIMC nonidealities and gauge their relative importance for consideration in future hardware designs. Functions for our standard evaluation process are provided in an open-source IBM Analog Hardware Acceleration Toolkit (AIHWKit)[50], enabling future studies on noise robustness for AIMC to build seamlessly upon our work.

We find that various DNNs and AI workloads−ranging across image classification using CNNs, text-prediction and speech-to-text conversion using recurrent neural networks (RNNs), and natural language processing using transformer networks−can actually be robustly deployed on AIMC given proper HWA training. We show iso-accuracy inference results (within 1% of the FP reference) using hardware-calibrated PCM models, for five out of the eleven AI workloads tested, even after 1 h (or more) of conductance drift.

gHowever, precision requirements are heterogeneous, and not all architectures reach this iso-accuracy target easily, even after extensive HWA training, pinpointing the need for continued device improvement. We find that CNNs are typically much less robust to various nonidealities and design choices of AIMC hardware. Interestingly, RNNs−already well-suited for AIMC given their large, dense MVMs[51]− also seem to be the most robust to the finite signal-to-noise ratio (SNR) of AIMC hardware. We further show that among various nonidealities tested, the sensitivity to additive system noise at the output of each crossbar array is the most critical for achieving good accuracy.

## Results
### Analog IMC standard MVM model
Our standard AIMC crossbar model (see Figs. 2 and 3, and Eqs. (1) and (2) in "Methods") encapsulates the critical nonidealities incurred during MVM operations, including the fixed dynamic ranges of physical inputs (limited by maximum pulse duration), weights (limited by maximum conductance), and outputs (limited by maximum output current). The nonideal MVM is a combination of digital computations close to the crossbar periphery, namely adjustable input scale $\alpha$ and column-wise output scales $\gamma_i$ and biases $\beta_i$, as well as fixed-range ADC and DAC quantizations:

$$\widetilde{y}_i = \beta_i + \alpha\gamma_i \, \mathrm{quant}^q_{b\,\mathrm{out}}^{\mathrm{out}} \left( \breve{\mathbf{F}}_i \left( \mathrm{quant}^q_1{}^{\mathrm{in}} (\mathbf{x}/\alpha) \right) \right), \tag{1}$$

where $\breve{\mathbf{F}}$ is an operator that describes the nonideal multiplication with the resisitve elements and accumulation of the crossbar currents, and $\mathrm{quant}^q_b(\cdot)$ indicates $q$ quantization steps in the range $-b, \ldots, b$ (with clipping; see Eq. (5)).

Thus, as illustrated in Fig. 2, digital FP inputs $x_i$ are scaled by a scalar $\alpha$, quantized in a fixed range (by the DAC), and then subject to the nonideal analog computation with noisy weights constrained by a fixed weight range (gray bell curves), as well as an additive system noise (blue bell curves). The (noisy) outputs of the analog crossbar are then digitized again by parallel ADC in a fixed output range, and finally re-scaled and shifted by the combined digital FP scales $\gamma_i\alpha$, and offsets $\beta_i$, respectively.

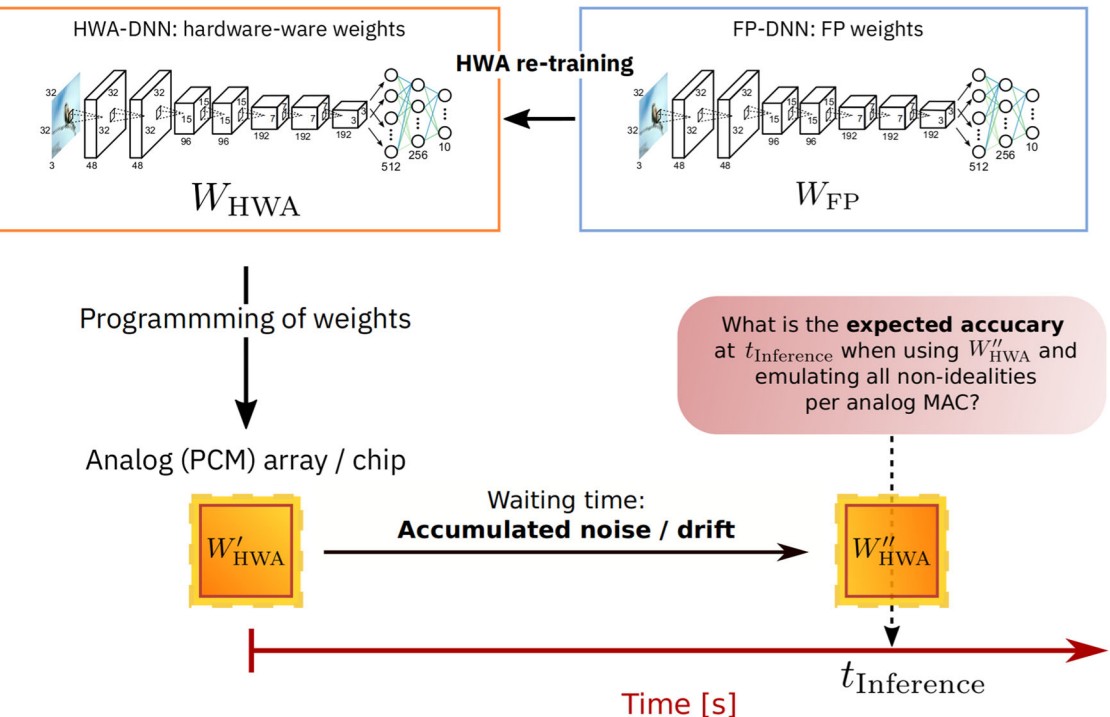

**Fig. 1 | Illustration of the HWA training approach.** In our hardware-aware (HWA) training setup, DNNs are first trained in 32-bit floating point (FP$_{32}$), then retrained in a hardware-aware manner, by adding nonidealities and noise sources into the forward path and using SGD to improve the robustness to such generic non-idealities. HWA training is only performed once—no specific device or chip characteristics, such as failure maps, are taken into account during HWA training, so resulting models remain widely deployable. This HWA-trained model is then programmed onto AIMC multiple times (here in simulation) and DNN accuracy is evaluated over time, taking into account conductance drift of PCM devices and read noise accumulation[31].

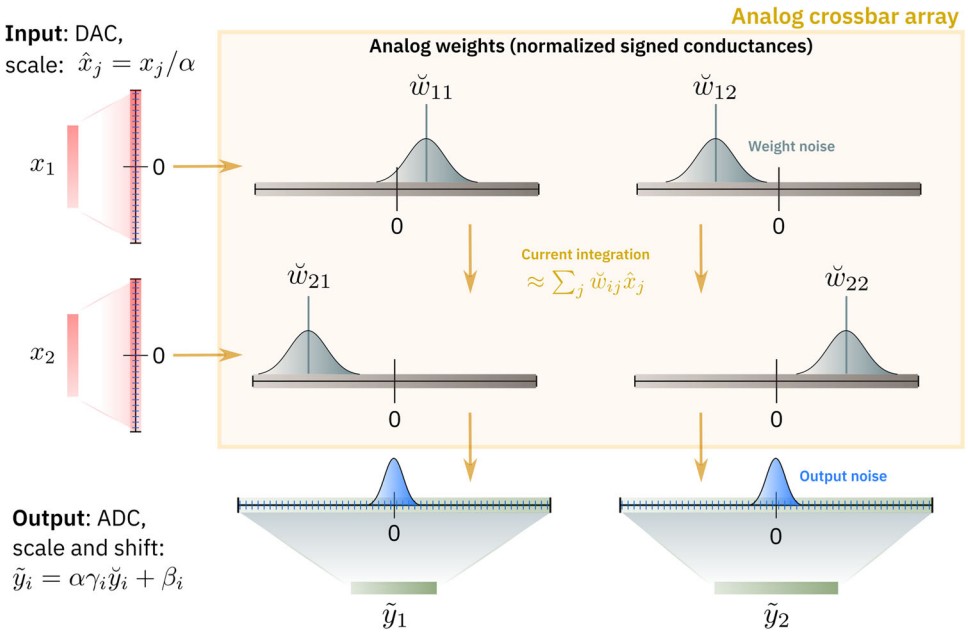

**Fig. 2 | Illustration of the AIMC crossbar-model abstraction.** Our analog in-memory computing (AIMC) crossbar model (using the nonideal matrix-vector multiplication (MVM) of Eqs. (1) and (2) together with hardware-calibrated PCM conductance noise and drift Eqs. (7)–(14) in "Methods") assumes that each array or "tile" approximates MVM $\tilde{\mathbf{y}} \approx W\mathbf{x}$, where digital inputs are converted with a digital-to-analog converter (DAC) to voltages, and current is integrated while weights are represented as conductances. Analog outputs are converted back from physical units to floating point (FP), using column-wise parallel analog-to-digital converters (ADCs), output scale vector $\boldsymbol{\gamma}$ and bias vector $\boldsymbol{\beta}$. Analog weight, input, and output ranges remain fixed; digital scales are used to map the FP weight values to the analog weights (ie. normalized conductances) of the crossbar and scale the ADC output ticks per column appropriately for subsequent (digital) layers. Negative weights are programmed onto a different conductance for current subtraction in the evaluation phase, and output noise is fully represented (see Eqs. (1) and (2)).

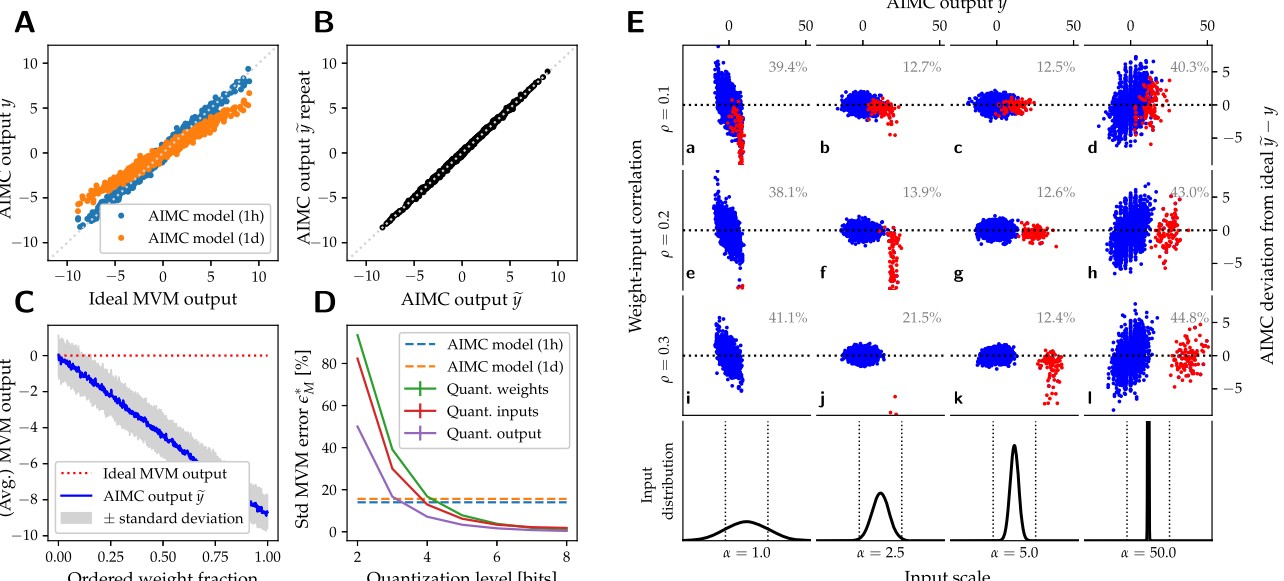

**Fig. 3 | Nonidealities of the AIMC crossbar model. A** Correlations between analog in-memory computing (AIMC) outputs—for matrix-vector multiplications (MVMs) performed between Gaussian random weight matrices and uniform random inputs —and the ideal (FP$_{32}$) expected results reveal significant deviations, due primarily to weight-programming errors and PCM conductance drift (shown here without any mean-drift compensation[36]). **B** Short-term noise sources induce cycle-to-cycle noise for repeated MVM calculations even with the same programmed weight matrix. **C** "IR-drops" due to finite wire resistance result from input-position dependency of the accumulated AIMC column-currents. An expected 0 output—the correct result when a linearly-graded weight matrix (ranging from −1 to 1 in order) is read with a constant input on all rows—can actually deviate drastically due to the degree of ordering of weights here shown from 0 (completely unordered, typical case) to 1 (fully ordered, extreme case). **D** The MVM error $\epsilon_M^*$ Eq. (20) of our

standard PCM-based AIMC inference model (Fig. 2; dotted lines) ≈15% roughly corresponds to fixed-point quantized digital (solid lines) at -4 bits. **E** Correlations of MVM deviation, $\tilde{y}_i - y_i$, vs. desired MVM output $y_i = w_{ij}x_j$ illustrate the importance of proper input scaling $\alpha$, for $w_{ij} \sim \mathcal{N}(0, 0.246)$ and $x_j \sim \mathcal{N}(0,1)$. Red dots mimic the weight-to-activation correlations that SGD learning will produce, using $\tilde{x}_j = \rho w_{kj} + (1 - \rho)x_j$, while blue dots represent the uncorrelated component for comparison, ($\rho = 0.0$). Low $\alpha = 1$ leads to input clipping (a, e, i) and $\epsilon_M^*$ exceeding 35% (gray text). Intermediate $\alpha$ values can still lead to saturated outputs for correlated inputs, even without input clipping (f, j); excessive $\alpha$ values reduce clipping but increase $\epsilon_M^*$ dramatically (d, h, l). We optimize $\alpha$ during hardware-aware (HWA) training, then keep it fixed during AIMC inference, minimizing $\epsilon_M^*$ regardless of input correlation (c, g, k).

The digital input scale $\alpha$ is optimized for each crossbar during HWA training, then held fixed for inference. Such optimization avoids issues created if $\alpha$ is chosen poorly (see Fig. 3E). Similarly, optimized scales ($\gamma_i$) and offsets ($\beta_i$) map ADC-counts of each output column to MVM outputs $\tilde{y}_i$ (see Eq. (1)) that can be passed to subsequent digital compute for auxiliary operations (activation functions, etc.)[51] (see also Supplementary Notes B.3 for an expanded discussion).

We further assume a number of nonidealities so that the analog MVM $\check{y} = \check{F}(\check{x})$ can be described mathematically as follows (with normal random variables $\xi_i, \xi_{ij} \sim \mathcal{N}(0,1)$):

$$\check{y}_i = \sigma^{out}\xi_i + f_i^{NL}\left(\Delta\check{y}_i^{IR\text{-}drop} + \sum_j \left(\check{w}_{ij}(t_{eval}) + \sigma^W\xi_{ij}\right)\check{x}_j\right). \quad (2)$$

Thus, our analog MVM model includes programming errors and drift ($\check{w}_{ij}(t_{eval})$; Fig. 3A), IR-drops within the array $\Delta\check{y}_i^{IR\text{-}drop}$ (Fig. 3C), short-term weight-dependent read noise ($\sigma^W = \sigma^W(\check{w}_{ij}(t_{eval}))$) and system noise ($\sigma^{out}$; Fig. 3B). We mainly investigate the situation where all weight-related parameters have been carefully calibrated to existing PCM hardware[31], however, the model can be adapted to other memory technologies as well (see Supplementary Notes B.2). Quantization levels (8-bit, ref. 44) and system noise (on the order of the ADC bin-width) are set to reasonable values by default, however, we will also explore their impact in a sensitivity analysis. The additional point-wise output nonlinearity $f_i^{NL}$ is assumed S-shaped in the sensitivity analysis only and otherwise omitted. For a more detailed discussion of the individual nonidealties, see "AIMC standardized evaluation model". All

parameter settings of the AIMC crossbar model are summarized in Supplementary Table 1.

We quantify MVM errors in computing $\tilde{y}$ with respect to the ideal outcome $y$ through $\epsilon_M$, the ratio of the $l_2$-norm of the deviation ($y - \tilde{y}$) relative to the $l_2$-norm of the ideal outcome $y$ (see Eq. (20)). Figure 3D shows that, even after including PCM drift, the effective MVM error of our standard AIMC crossbar model roughly corresponds to 4-bit fixed-point quantization of weights or inputs.

### DNN accuracy impact when directly using AIMC

To test the effect of AIMC nonidealities on a variety of AI workloads, we consider 11 medium- to large-scale DNNs of various topologies as a benchmark set (see Table 1). These cover a wide spectrum of target-applications (image classification, natural language processing, speech-to-text), network topologies (convolutional, recurrent, transformer with attention), model sizes (from 0.3M to 108M parameters), crossbar utilization (from 4% to 86%), total number of MVMs per data input (from 2.2K to 240K), MVM sizes (from 0.1G to 96.5G flops), average weight-matrix reuse-factor per data input (from 17 to 1285), and network depth (up to 121 layers). Our benchmark set thus covers a wide variety of network topologies and challenges for AIMC.

For comparison, we first directly map weights produced by standard stochastic gradient descent (SGD)-training in FP$_{32}$ onto our standard AIMC crossbar model and evaluate the resulting test error, to measure the accuracy drop (with respect to the FP$_{32}$ model) due to all the various AIMC nonidealities. Output scales $\gamma_i$ are initially estimated according to the absolute maximum weight value for each column (see Eq. (21)); having individual scales per column available in the chip

## Table 1 | Properties of the benchmark set of 11 DNNs

| DNN | Type | # par. | # mapped | # tiles | util. | # MVM | ⟨reuse⟩ | Flops |
|---|---|---|---|---|---|---|---|---|
| ResNet-32 CF10 | C | 0.36M | 0.36M | 34 | 4.0% | 14K | 435. | 0.1G |
| WideResNet-16 CF100 | C | 11M | 11M | 68 | 61.7% | 19K | 286. | 3.1G |
| ResNet-18' ImNet | C | 11.7M | 11.2M | 75 | 56.8% | 40K | 533. | 3.4G |
| ResNet-50' ImNet | C | 25.6M | 23.4M | 149 | 60.0% | 74K | 497. | 7.9G |
| DenseNet-121' ImNet | C | 8M | 6.9M | 266 | 9.8% | 143K | 537. | 5.4G |
| WideResNet-50' ImNet | C | 69M | 66.8M | 296 | 86.0% | 101K | 342. | 22.6G |
| BERT-base MRPC | T | 108M | 85M | 486 | 67.1% | 61K | 126. | 21.8G |
| Albert-base MRPC | T | 12M | 7.8M | 48 | 61.7% | 61K | 1285. | 21.8G |
| Speech☐SWB300 | L | 30M | 30M | 153 | 74.8% | 2.5K | 17. | 0.9G |
| LSTM PTB | L | 19.8M | 13.3M | 88 | 57.5% | 2.2K | 26. | 0.7G |
| RNN-T | L | 57M | 57M | 304 | 71.6% | 240K | 790.* | 96.5G |

A wide range of DNNs topologies (Type) and sizes are studied, including CNNs (C), LSTMs (L), and transformers (T). For each DNN model and dataset, model size is quantified by number of parameters (# par.); number of parameters mapped to analog crossbars (# mapped); number of 512 × 512 crossbars (# tiles) needed for naive mapping (each weight matrix gets at least 1 tile); overall utilization of the devices within the tiles (util.); total number of MVMs per input data (# MVM); average tile-reuse for one input data (⟨reuse⟩; *for maximal input length in dataset); and the number of $FP_{32}$ operations in the mapped MVM for one input data (FLOPS); 'first conv-layer and last FC layer in $FP_{32}$; ☐additional hidden Markov model used as decoder.

## Table 2 | Inference of floating-point (FP)-trained DNNs

| Direct mapping | Test error in % | | | | | Normalized acc. in % | | |
|---|---|---|---|---|---|---|---|---|
| DNN | $FP_{32}$ | 1 second | 1 hour | 1 day | 1 year | $\mathcal{A}_*^{1h}$ | $\mathcal{A}_*^{1d}$ | $\mathcal{A}_*^{1y}$ |
| ResNet-32 CF10 | 5.80 | 12.25 ± 0.29 | 13.14 ± 0.41 | 13.64 ± 0.34 | 18.49 ± 0.62 | 91.3 | 90.7 | 84.9 |
| WideResNet-16 CF100 | 20.00 | 24.06 ± 0.16 | 25.11 ± 0.20 | 25.74 ± 0.31 | 30.58 ± 0.39 | 93.5 | 92.7 | 86.6 |
| ResNet-18' ImNet | 30.50 | 36.56 ± 0.17 | 37.98 ± 0.16 | 40.11 ± 0.34 | 47.61 ± 0.69 | 89.2 | 86.2 | 75.4 |
| ResNet-50' ImNet | 23.87 | 35.28 ± 0.17 | 36.51 ± 0.28 | 37.77 ± 0.23 | 46.25 ± 0.61 | 83.4 | 81.7 | 70.6 |
| DenseNet-121' ImNet | 25.57 | 35.25 ± 0.29 | 35.59 ± 0.34 | 38.82 ± 0.30 | 53.73 ± 0.72 | 86.5 | 82.2 | 62.1 |
| WideResNet-50' ImNet | 21.53 | 33.41 ± 0.19 | 34.14 ± 0.13 | 36.59 ± 0.22 | 46.54 ± 0.33 | 83.9 | 80.8 | 68.1 |
| BERT-base MRPC | 14.60 | 21.77 ± 0.21 | 22.87 ± 0.20 | 23.39 ± 0.26 | 28.01 ± 0.18 | 83.9 | 82.9 | 73.9 |
| Albert-base MRPC | 15.08 | 32.00 ± 0.00 | 32.00 ± 0.00 | 31.25 ± 0.00 | 31.50 ± 0.00 | 66.8 | 68.2 | 67.8 |
| Speech☐SWB300 | 14.05 | 21.40 ± 0.03 | 15.43 ± 0.02 | 14.77 ± 0.02 | 14.78 ± 0.02 | 98.4 | **99.2** | **99.1** |
| LSTM PTB | 72.90 | 72.98 ± 0.01 | 73.11 ± 0.01 | 73.27 ± 0.01 | 73.52 ± 0.02 | **99.2** | 98.6 | 97.7 |
| RNN-T SWB300 | 11.80 | 18.90 ± 0.34 | 12.33 ± 0.02 | 12.35 ± 0.02 | 12.86 ± 0.05 | **99.4** | **99.4** | 98.8 |

Inference results using analog in-memory computing (AIMC) for the 11 benchmark DNNs when deployed directly without any weight retraining. Test errors in % ± standard error of mean (across 24 inference repeats; italic font) are shown after 1 hour and 1 year of PCM drift (center columns) and compared to the original $FP_{32}$ test error (leftmost column). Digital parameters needed for the AIMC crossbar model are estimated by initial conductance mapping according to Eq. (21) and training briefly with the AIMC MVM in the forward pass (1000 batches), but without touching the directly mapped analog weights. This helps adjust statistics of each batch norm to the new output distributions caused by the AIMC MVMs. During these 1000 batches, we estimate α by averaging the maximal absolute inputs for each batch during the first 500 batches, and then allow SGD to tune it further during the second half of the brief digital-parameter-only HWA training. Right-hand columns show normalized accuracy values after 1 hour ($\mathcal{A}_*^{1h}$) and 1 year ($\mathcal{A}_*^{1y}$), as scaled to the range between the $FP_{32}$ reference test error and the test error obtained by random guessing. Except for a respectable result for the RNNs (where in some cases longer PCM drift counter-intuitively improves accuracy because the reduction of the weight values better meets the fixed output range constraints), all other models fail to achieve values close to iso-accuracy, as defined by >99% in this normalized accuracy and indicated in bold font. Note that for BERT and Albert only one GLUE task (MRPC) is used here; 'first conv-layer and last FC layer in FP32; ☐additional hidden Markov model used as decoder.

design is crucial, see Supplementary Table 6 for direct mapping results if only a single output scale $\gamma$ is available. To adjust the digital parameters of our standard AIMC crossbar model for these directly mapped-from-software weights, we use our HWA training flow—but without any weight noise injection, with weight learning rates set to zero, and for only 1000 batches. As expected, such "direct" mapping of DNNs onto AIMC, without any additional retraining of the weights, generally results in significant increases in test error (accuracy drop) in comparison to the floating-point reference (Table 2).

Direct comparison of accuracy values between DNNs is complicated by the fact that these various AI tasks exhibit different worst-case (random guessing) and best-case (well-trained DNN model) accuracies. To quantify and compare accuracy drop across different topologies, we therefore define a normalized relative accuracy $\mathcal{A}^{1h}$, which re-scales the AIMC test error $\epsilon_{test}^{1h}$ (at 1 h PCM drift) by the distance between the original $FP_{32}$ test error and the "chance" test error from random guessing, as follows:

$$\mathcal{A}^{1h} = 1 - \frac{\epsilon_{test}^{1h} - \epsilon_{test}^{FP}}{\epsilon_{chance} - \epsilon_{test}^{FP}}. \tag{3}$$

Thus a value of $\mathcal{A}^{1h} = 100\%$ means that the AIMC DNN achieves the same accuracy as the $FP_{32}$ reference model (no accuracy drop at all), while a value of $\mathcal{A}^{1h} = 0\%$ implies that the AIMC crossbar model is so inaccurate that it is indistinguishable from random guessing.

Ideally, deploying a given DNN in an AIMC system should have no impact on model accuracy. We define our iso-accuracy target as $\mathcal{A}_*^{1h} > 99\%$, allowing less than a 1% drop in accuracy, as judged relative to the distance between the $FP_{32}$ reference accuracy and the chance (random guessing) accuracy floor. Table 2 shows that direct AIMC mapping fails to achieve this iso-accuracy target for almost all of the DNNs tested, establishing both the challenge posed by the non-idealities existing in AIMC (as compactly encapsulated by our standard crossbar model, Figs. 2 and 3), as well as the need for HWA training methods that can greatly improve the robustness and reduce these accuracy drops.

### HWA training improves AIMC accuracy for all DNNs

Building on previous approaches (see refs. 38,40,41), we set out to retrain these 11 DNNs in a hardware-aware (HWA) manner. In our methodology for HWA training followed by delayed inference (Fig. 1), each DNN is retrained with noise injection using SGD. But in contrast to earlier approaches, we incorporate a much more comprehensive and

realistic set of software-simulated AIMC nonidealities, including dynamic-range limitations, weight-programming errors, PCM drift and system noise. Once a given DNN is trained and mapped to AIMC, the inference is then gauged for noise and drift at various delays (1 s, 1 h, 1 day, and 1 year) after programming the weights into the crossbar arrays. We also introduce a set of AIMC characteristics, including input, output, and weight scales (see Fig. 3 and "Methods"), and introduce an approach for optimizing these scaling factors during HWA training for use during inference (see Sec. B.1).

As shown in Table 3, our HWA training approach significantly improves achievable accuracy for AIMC across the full set of benchmark DNN results. The normalized accuracies (relative to the FP$_{32}$ model) at 1 h after programming are all higher than 97% ($\mathcal{A}_*^{1h}$, toward right edge of Table 3). This represents a significant improvement over "direct" weight mapping without retraining shown earlier (Table 2), while establishing a state-of-the-art in HWA training, as revealed by detailed comparisons on ResNet-32 with CIFAR10 (see Supplementary Table 2).

Table 3 indicates that five out of the 11 AI workloads can be trained to reach the $\mathcal{A}_*^{1h} > 99\%$ iso-accuracy target, including the BERT transformer model as well as all workloads based on long short-term memory (LSTMs) (last 3 rows, see "Type" column in Table 1). Most of the remaining workloads use CNNs and exhibit more-pronounced accuracy drops of up to 2.8% on AIMC, although one CNN does reach iso-accuracy (WideResNet-16 on CIFAR100).

For some DNNs, we find that the regularization effect of the added AIMC nonidealities allows HWA training to actually improve the attainable accuracy (compare test errors at 1 s after programming for WideResNet-16 and BERT). Both RNNs and transformers are quite robust when subject to PCM conductance drift over longer periods as well. The rightmost column of Table 3 shows the long-term relative accuracy of the DNNs, $\mathcal{A}_*^{1y}$, for an hypothetical 1 year after programming without weight refresh.

While the RNNs and transformers remain near iso-accuracy over time, larger CNNs with higher resolution ImageNet inputs show the largest drop in accuracy. The deep DenseNet-121 (121 layers), as well as the large WideResNet-50 (69M parameters), and the Albert transformer (with layer re-usage) models are the most challenging for AIMC. That said, the resiliency to long-term drift is greatly improved by HWA training as compared to "direct" deployment without retraining. For

instance, the HWA-trained models for both the Speech-SWB300 and LSTM-PTB models remain iso-accurate out to a year, unlike the directly mapped models (Table 2).

In general, we find that CNNs are more difficult to train to iso-accuracy for AIMC deployment compared to RNNs and transformers. In terms of AIMC workload execution latency and system mapping[51], CNNs are already less well-suited for resistive crossbar arrays due to the uneven temporal reuse between layers and spatial under-utilization of the large analog tiles by the small kernel matrices (see Table 1), although some optimization and mapping tricks[52] are available. Our results here indicate that AIMC noise-robustness issues will pose additional challenges when implementing CNNs onto AIMC systems.

**Sensitivity of HWA-trained models to various AIMC nonidealities**

To determine which nonidealities are particularly problematic for analog inference across DNNs, we "stress test" our HWA-trained models. For each individual nonideality, such as PCM programming error or IR-drop, we vary its strength and evaluate the resulting inference accuracy across DNNs using our base HWA-trained model. Our standard AIMC MVM model exhibits $\epsilon_M^* \approx 15\%$ (see Fig. 3 and Eq. (20)), but combines many nonidealities. To estimate the relative accuracy impact due to each individual nonideality, we boost only that parameter value until MVM error increases to $\epsilon_M^* = 20\%$, and then re-measure DNN accuracy.

Even at constant MVM error, each parameter changes a different aspect of the AIMC compute. For instance, output noise is applied at each MVM, whereas PCM programming errors are only applied during programming and then persist throughout inference. Other non-idealities such as IR-drop or ADC "S-shaped" nonlinearity change the shape of the MVM deviations (Fig. 4A), causing large outputs to incur very significant MVM error. As a result, even at an identical average MVM error of $\epsilon_M^* = 20\%$, the impact on DNN accuracy can be much more pronounced. Such nonidealities are particularly detrimental for DNN inference, and thus deserve additional attention in future hardware designs or HWA training methods.

To gauge the relative impact of each individually boosted non-ideality parameter, Fig. 4B shows the loss in normalized accuracy ($\mathcal{A}^{1h}$), defined not with respect to the FP$_{32}$ model error ($\mathcal{A}_*^{1h}$ Eq. (3)), but with respect to our standard AIMC crossbar model (at 1-h drift). A value of

**Table 3 | Inference of HWA-trained DNNs**

| HWA training | Test error in % | | | | | Normalized acc. in % | | |
|---|---|---|---|---|---|---|---|---|
| DNN | FP$_{32}$ | 1 second | 1 hour | 1 day | 1 year | $\mathcal{A}_*^{1h}$ | $\mathcal{A}_*^{1d}$ | $\mathcal{A}_*^{1y}$ |
| ResNet-32 CF10 | 5.80 | 6.73 ± 0.02 | 6.99 ± 0.02 | 7.33 ± 0.03 | 8.55 ± 0.09 | 98.6 | 98.2 | 96.7 |
| WideResNet-16 CF100 | 20.00 | 19.61 ± 0.02 | 19.78 ± 0.02 | 20.10 ± 0.02 | 21.12 ± 0.03 | **100.3** | **99.9** | 98.6 |
| ResNet-18† ImNet | 30.50 | 31.28 ± 0.02 | 31.59 ± 0.02 | 31.98 ± 0.03 | 33.43 ± 0.05 | 98.4 | 97.9 | 95.8 |
| ResNet-50† ImNet | 23.87 | 24.56 ± 0.01 | 24.83 ± 0.02 | 25.29 ± 0.03 | 27.21 ± 0.04 | 98.7 | 98.1 | 95.6 |
| DenseNet-121† ImNet | 25.57 | 26.46 ± 0.02 | 26.96 ± 0.03 | 27.67 ± 0.04 | 30.85 ± 0.08 | 98.1 | 97.2 | 92.9 |
| WideResNet-50† ImNet | 21.53 | 23.43 ± 0.02 | 23.76 ± 0.02 | 24.21 ± 0.03 | 26.71 ± 0.06 | 97.2 | 96.6 | 93.4 |
| BERT-base GLUE8 | 17.47 | 17.43 ± 0.09 | 17.55 ± 0.12 | 17.58 ± 0.12 | 17.99 ± 0.12 | **99.8** | **99.8** | 98.9 |
| Albert-base GLUE8 | 19.46 | 20.52 ± 0.18 | 20.45 ± 0.16 | 21.08 ± 0.18 | 22.18 ± 0.21 | 97.8 | 96.4 | 94.0 |
| Speech□SWB300 | 14.05 | 14.24 ± 0.01 | 14.24 ± 0.01 | 14.29 ± 0.02 | 14.42 ± 0.02 | **99.8** | **99.7** | **99.6** |
| LSTM PTB | 72.90 | 72.97 ± 0.00 | 73.00 ± 0.00 | 73.02 ± 0.00 | 73.10 ± 0.01 | **99.6** | **99.6** | **99.3** |
| RNN-T SWB300 | 11.80 | 12.22 ± 0.04 | 12.36 ± 0.02 | 12.42 ± 0.04 | 12.78 ± 0.04 | **99.4** | **99.3** | 98.9 |

Test error in % ± standard error of mean (across 15–25 inference repeats per training trial and up to three training trials; italic font) for DNN deployment on analog in-memory computing (AIMC) crossbars after hardware-aware (HWA) training. Rightmost two columns show the normalized accuracy, scaled between the FP reference and chance error, at 1 h, 1 day, and 1 year after weight programming. Note that PCM drift is a post-programming physical effect that is initially rapid but then slows down logarithmically in time[23]. This means that the multiplicative conductance changes induced by drift between 1 s and 1 h (time-since-programming increased 3600×), and between 1 h and 1 year (time-since-programming increased 8760×) are actually quite similar. HWA training hyper-parameters (injected noise strength, etc.) were chosen to produce the best average accuracy across the four widely spaced timepoints shown here. Other choices could be made to focus just on performance in either longer or shorter periods of drift. Models deemed iso-accurate ($\mathcal{A}_* > 99\%$) are marked in bold. BERT and Albert results are averaged across eight GLUE tasks, as evaluated on validation datasets; SWB300 results are averaged over two benchmark tasks; results for Speech-SWB300 and WideResNet-16–CIFAR100 use HWA with distilling.†first conv-layer and last FC layer in FP32; □additional hidden Markov model used as decoder.

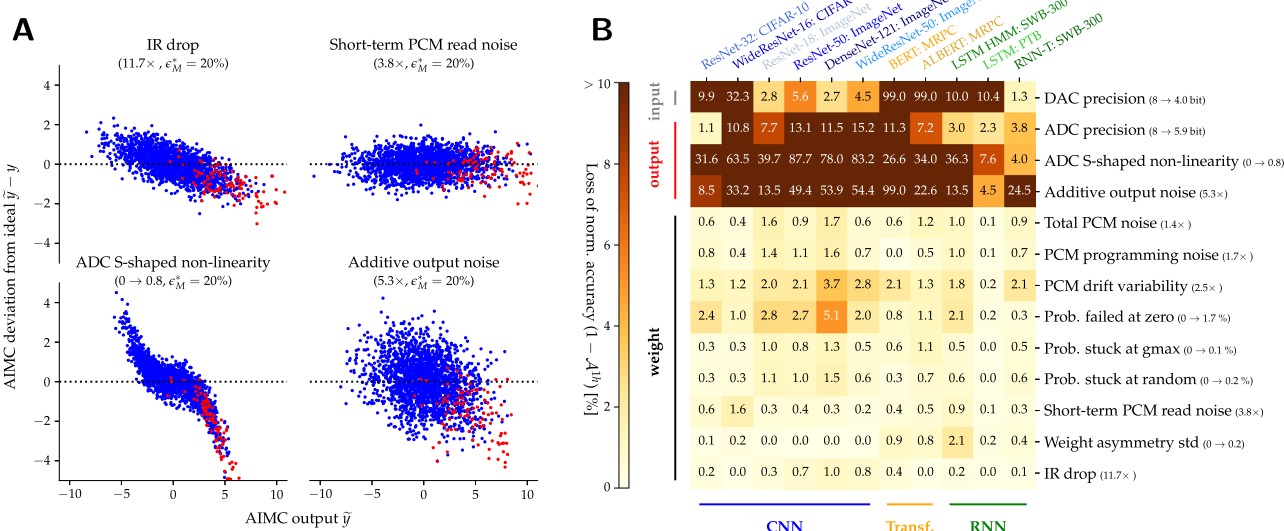

**Fig. 4 | Comparison of AIMC nonidealities.** Comparison of the relative impact of various analog in-memory computing (AIMC) nonidealities on DNN accuracy. **A** AIMC deviations ($\tilde{y} - y$) from the ideal matrix-vector multiplication (MVM) output ($y$) are shown, for uncorrelated (blue dots) and weakly correlated ($\rho = 0.05$) random activations (red dots), as a single nonideality is increased until standard MVM error reaches $\epsilon_M^* = 20\%$. All other parameters remain fixed to our standard crossbar model ($\epsilon_M^* = 15\%$, Fig. 3). For instance, IR-drop needs to be scaled 11.7 × to incur $\epsilon_M^* = 20\%$. Even at constant $\epsilon_M^* = 20\%$, MVM deviations are structured differently and thus the impact on DNN accuracy can vary significantly. **B** Grid shows loss in normalized accuracy ($\mathcal{A}^{1h}$) over the base HWA-trained model at 1 h after programming when boosting a given nonideality to $\epsilon_M^* = 20\%$. Thus 0% means no accuracy impact despite the amplified nonideality, whereas 100% means a drop to chance level. For the HMM, LSTM sensitivity is reported for a portion of the training set (instead of the benchmark set) directly on the LSTM output without the hidden Markov model to speed up computations. For the transformer models, only one GLUE task is evaluated (MRPC).

0% means that boosting this particular nonideality has no impact on accuracy, as compared to our standard AIMC crossbar model. A value of 100% means that simply boosting this nonideality to the same MVM error of $\epsilon_M^* = 20\%$ has degraded DNN accuracy to the level of random guessing.

Clearly, DNN accuracy reacts vastly differently to individual nonidealities. We observe that nonidealities that effectively add noise to the inputs or outputs—such as ADC and DAC resolution, additive output noise, and S-shaped nonlinearity of the ADC—have the largest impact on the DNN accuracy, as normalized to impact on average MVM error. CNNs are the most-sensitive DNN topology, while RNNs are the least-sensitive (in particular the PTB-LSTM network).

Nonidealities that mostly affect weight precision (all other nonidealities listed in Fig. 4B), have a much less severe impact on the DNN accuracy. In contrast to additive output noise, such weight-related nonidealities all scale with the input norm, and thus disappear when no inputs are given. Since it arises from large currents, IR-drop becomes negligible when either inputs or weights are reduced (in either amplitude or occurrence). Such weight-related nonidealities impact CNNs slightly more than RNNs or transformers. In particular, DenseNet-121 with small kernel matrices and a high tile reuse factor seems the most affected by weight disturbances. Figure 4 shows it is not enough to focus only on weight-related nonidealities, as most previous studies have done, when investigating AIMC.

We use this sensitivity analysis to assess additional nonidealities which our standard AIMC crossbar model assumes to be perfect. For instance, imperfect device yield—where some fraction of the weight conductances are "stuck" either at zero (PCM reset), at $\hat{g}_{max}$ (PCM set), or at some intermediate random value —is shown to have the same modest effect on DNN accuracy as other weight-related parameters. Weight asymmetry—a systematic difference in conductance for positive versus negative inputs such that $-w(-|x|) \neq w(|x|)$ – is shown to have only modest impact on DNN accuracy. Interestingly, RNNs and transformers are the models impacted by such polarity-dependent device response, since the ReLU activations used in CNNs cannot create negative inputs. Finally, systematic PCM programming errors—applied once to the conductance values and then remaining constant through repeated MVMs—are shown to have a slightly larger effect than the cycle-to-cycle short-term PCM read noise that gets redrawn for every MVM.

## AIMC robustness of DNN topologies
To extract the specific sensitivities of each individual DNN, we find the threshold value $x^*$ at which each nonideality degrades accuracy to $\mathcal{A}^{1h}(x) = 99\%$, with respect to the standard AIMC crossbar model. From scans of $\mathcal{A}^{1h}$ as each nonideality is increased (Fig. 5A), we use linear interpolation to identify $x^*$ from the intersection with the dotted line at $\mathcal{A}^{1h} = 99\%$.

The grid in Fig. 5B shows this threshold value $x^*$, for each nonideality and each DNN. For example, considering just total PCM noise, even small increases beyond the current hardware-calibrated values markedly degrade ResNet-18 ($x^* = 1.2 \times$ for $\mathcal{A}^{1h} = 99\%$), while LSTM-PTB is not affected until this particular nonideality is significantly larger ($x^* = 3.3 \times$). The colors ranging from red to green in Fig. 5 illustrate the relative sensitivity among the DNNs, obtained by scaling $x^*$ linearly between the minimal and maximal values across the 11 DNNs. For many of these nonidealities, yet again RNNs tend to be the most robust, followed by small CNNs on the CIFAR datasets.

Some nonideality parameters can be increased quite dramatically with respect to our standard AIMC crossbar-model baseline. For instance, DAC precision can be lowered from 8-bit to 6-bit without any retraining, with little accuracy impact across all DNNs—this could produce considerable energy savings and throughput improvement for AIMC designs. Also, IR-drop can be increased beyond the baseline before becoming problematic, and short-term weight noise could be up to 3 × larger, similarly informing future AIMC designs, both with and without PCM devices. While direct examination of Fig. 5 might suggest that IR-drop could be increased by 10 × without issue, note that the assumptions inherent in our IR-drop calculations, concerning average

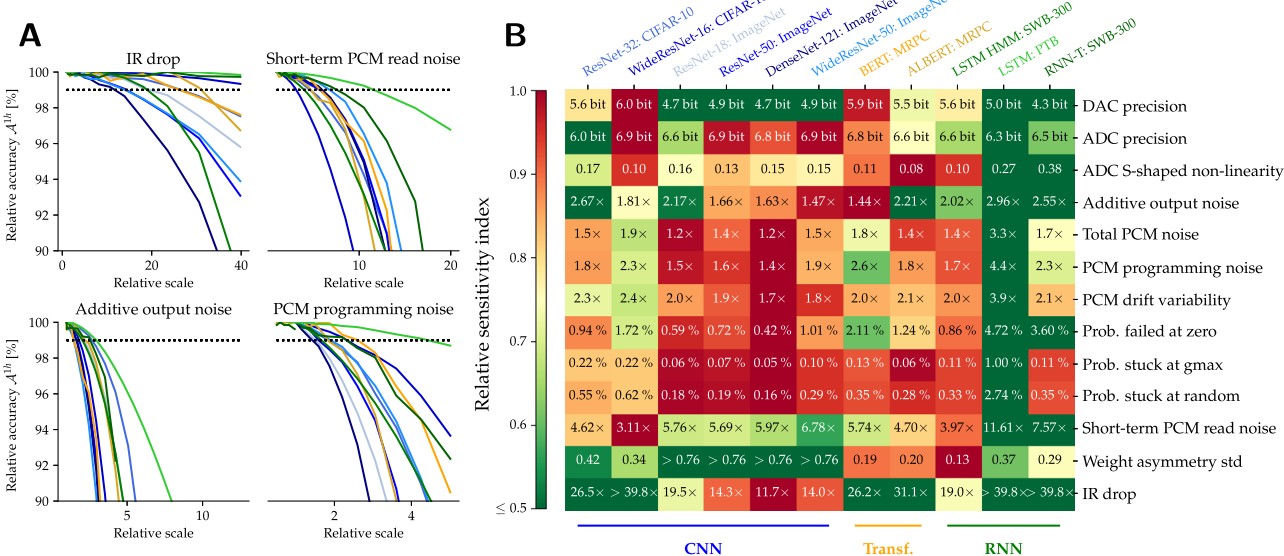

**Fig. 5 | Specifications of AIMC nonidealities.** Tolerances of individual analog in-memory computing (AIMC) nonidealities across DNNs. **A** As a single nonideality parameter is increased from the our standard setting, accuracy $\mathcal{A}^{1h}$ eventually drops to 99% (compared to accuracy of the standard AIMC crossbar model). Four nonidealities are shown, with DNN line-colors matching the text-label color in (**B**). **B** Grid shows $x^*$, the threshold value at which that particular nonideality produces $\mathcal{A}^{1h} = 99\%$ (DNN's curve crosses dotted line in (**A**)). For instance, reducing DAC precision from 8-bit down to 5-bit, while maintaining all other parameters from the standard AIMC crossbar model, causes exactly 1% additional accuracy loss in the LSTM-PTB model. Text-label colors at the top match the lines in (**A**); grid colors reflect relative sensitivity index $r_s = \frac{x^* - \min x^*}{\max x^* - \min x^*}$, with min and max values taken across all DNNs. $r_s = 1$ (red) indicates the most-sensitive and $r_s = 0$ (green) the least-sensitive DNN. RNNs are generally observed to be more robust to AIMC non-idealities than CNNs, even with the limited hyper-parameter tuning available for RNN-T due to its large number of MVM FLOPS and parameters.

rather than instantaneous currents, imply a small safety margin of perhaps 3× (see "Methods").

We also estimated the effect of imperfect PCM device yield. Even the least robust model can tolerate 0.42% failed-at-zero devices (stuck in the reset state, at random locations), rising to 3–4% for some of the RNNs. However, DNN accuracies are more sensitive to devices stuck either at random intermediate conductance values or at $\hat{g}_{max}$ (in the set state). As few as 0.05% of such failed devices would already cause a noticeable accuracy drop in some large CNNs. However, our analysis only assumes one pair of conductances per weight —since many existing AIMC designs provide multiple pairs of PCM devices per weight[44,47], such additional redundancy can potentially counteract such stringent device yield requirements.

**Impact of weight distributions on AIMC MVM fidelity**

The MVM error of each AIMC crossbar is affected by the shape of the weight distributions in interesting ways. While weight-clipping might seem disadvantageous, directly programming a very "long-tailed" weight distribution by mapping its largest outlying weight value to $\hat{g}_{max}$ can cause even larger problems. Such mappings tend to produce low average output currents which fail to employ the available ADC range, leading to larger MVM errors thanks to ADC quantization, output noise, and other nonidealities that remain stubbornly independent of the reduced output signal levels.

To show this effect, we calculate the MVM error for different arbitrarily-constructed weight distribution shapes, obtained by sampling the generalized normal distribution,

$$p(x|\mu,\alpha,\beta) = \frac{\beta}{2\alpha\Gamma(1/\beta)} \, e^{-(|x-\mu|/\alpha)^\beta}, \quad (4)$$

where we use $\alpha = 1$ and $\mu = 0$. As $\beta$ increases, this distribution becomes more compact, moving through the Laplace ($\beta = 1$) and normal distributions ($\beta = 2$) along the way (see red curves above Fig. 6A). Figure 6A shows the MVM error $\epsilon_M$ at 1-h drift, for weight values sampled

from Eq. (4) as $\beta$ increases from long-tailed ($\beta \leq 1$) to compact (high $\beta$) weight distributions. Here we map weights directly to conductance values, with the maximum weight assigned to $\hat{g}_{max}$; inputs are uniformly distributed between (−1, 1). MVM error increases rapidly for longer-tailed distributions ($\beta \leq 1$).

One simple measure of a distribution's shape is the kurtosis, obtained by dividing the fourth moment ($\langle(x-\mu)^4\rangle$) of the distribution by its variance squared ($[\langle(x-\mu)^2\rangle]^2$). In the plots and the remainder of this section, we use the excess kurtosis—defined as the kurtosis minus 3, so that its value is 0 for normal distributions. Since kurtosis increases for long-tailed distributions, we find that lower kurtosis—and thus more compact weight distributions—means lower MVM error (Fig. 6B).

Fortunately, our HWA training and conductance mapping approach tends to inherently produce more compact conductance distributions, for several different reasons. First, the individual digital scales $\gamma_i$ available for each MVM output (see Eq. (1)) are initialized to scale conductances by the absolute maximal value of each weight-matrix-column rather than by the overall maximum across the entire weight matrix. With each column individually scaled, the overall conductance distribution becomes more compact than the original weight distribution. During HWA training, these digital scales are optimized—which may lead the system to choose to clip some output columns—and any large weight deviations and outliers created during training are also clipped. Finally, since the AIMC nonidealities cause large weights and outputs to increase the errors that SGD is attempting to correct, HWA training should be expected to drive towards more compact weight distributions during retraining.

Indeed, we find that our HWA training and mapping scheme greatly increases the compactness of the conductance distributions for each layer, as indicated by the kurtosis values shown for our 11 DNN models in Fig. 6C. Hashed bars show kurtosis for direct mapping of the $FP_{32}$ model without HWA training, using a single global digital scale factor per layer. Solid bars illustrate that our column-wise-scaled and HWA-trained models get mapped into conductance distributions that

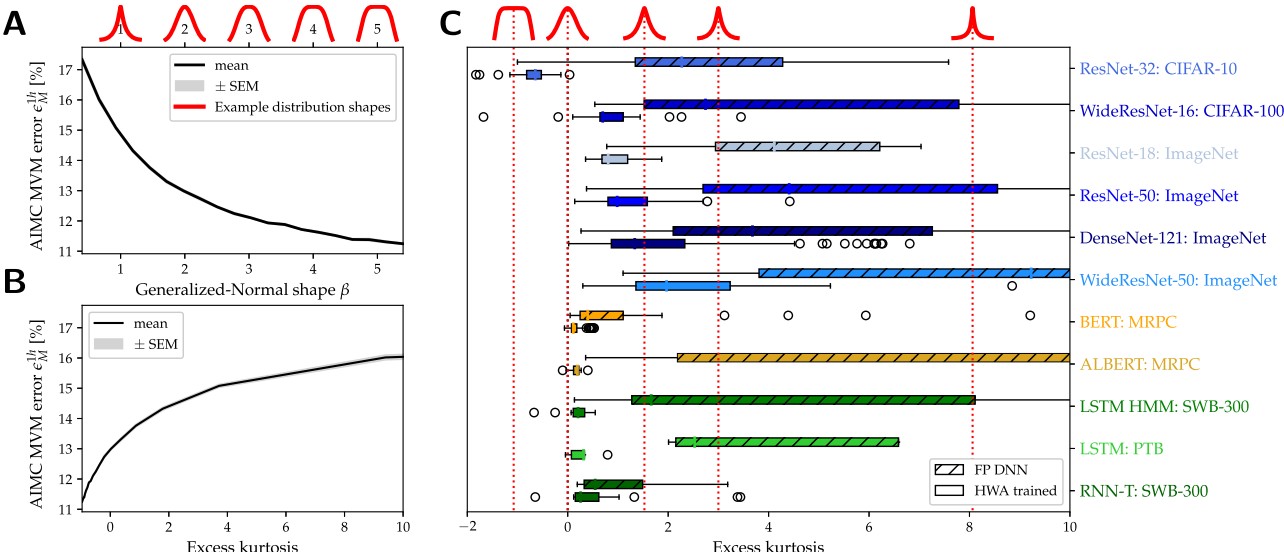

**Fig. 6 | Compactness of conductance values distributions.** The hardware-aware (HWA) training reduces matrix-vector multiplication (MVM) error by creating more compact conductance distributions. **A** MVM error decreases as constructed conductance distributions, produced by a generalized normal distribution Eq. (4), are made more compact by increasing $\beta$. Example distributions in red at top show $\beta = 1$ (Laplace distribution), $\beta = 2$ (normal distribution), and even more compact distributions for higher $\beta$. "SEM" indicates standard error of the mean. **B** Data from (**A**) is replotted as a function of the (excess) kurtosis of the distribution. According to the definition of excess kurtosis, a normal distribution (that is $\beta = 2$ in A) has a value of 0, and positive or negative values for longer tail distributions (i.e., $\beta < 2$) or more compact distributions (ie. $\beta > 2$), respectively. Note that longer tail distributions (large kurtosis) lead to higher MVM error, while more compact distributions (lower kurtosis) reduce MVM error (**C**) Kurtosis of the conductance values per layer, comparing HWA-trained models (solid bars), to FP32 weight data scaled by the overall absolute maximum weight (hashed bars). Column-wise scaling, and the tuning of both weights and scaling parameters during HWA training, help lead to significantly more compact distributions with smaller kurtosis values. Box plot shows first quartile and third quartile with a line at the median. The whiskers extend 1.5× the interquartile range. Outlier points are those past the end of the whiskers.

are significantly more compact, which helps reduce both MVM and DNN error.

## Improving AIMC fidelity of selected layers to reach iso-accuracy in large CNNs

Our results show that larger CNNs, particularly those using the ImageNet database, are the most challenging for AIMC. Even with HWA training, our standard AIMC crossbar model cannot achieve iso-accuracy for these DNN models (Table 3). Clearly, the fidelity of the MVMs must be further improved, either through better materials or through hardware design choices. For instance, designers could dedicate multiple conductance pairs per weight[53] to reduce PCM programming errors, but at the cost of larger tile area and energy. Or designers could average the results from multiple passes through the tile to reduce the effects of cycle-to-cycle PCM read and additive output noise, but at significant cost to latency, throughput, and energy efficiency. Given these unpleasant tradeoffs, such approaches should be used as infrequently as possible, ideally only on a small set of DNN layers that really require these extra resources, which can then allow the entire model to achieve iso-accuracy.

Thus, we need to determine which of the layers in ImageNet CNNs are the most-sensitive to AIMC nonidealities, and then assess whether improving just a small subset of these layers would have sufficient impact. To do this, we sequentially introduce AIMC nonidealities at each layer of the HWA-trained DNNs individually, while turning off all nonidealities in all other layers (using FP32 operations on their HWA-trained weight matrices). By repeating this process over the $L$ layers with different overall PCM noise settings, we can determine the sensitivity and relative importance of single layers.

We first rank the layers according to accuracy impact for each DNN by exposing each layer to significant PCM noise with all other layers exempted from noise (Fig. 7A). Then, in order from most- to least-sensitive layer, we introduce this noise-exemption into multiple layers (Fig. 7B), causing normalized accuracy at 1-h drift $\mathcal{A}_*^{1h}$ with

respect to the FP32 model to increase as more and more model parameters are made noise-exempt (Fig. 7C). Eventually the 99% iso-accuracy is achieved (dashed horizontal line) and then exceeded for most of these models. For Fig. 7A, the one layer being assessed sees 15 × the usual PCM noise; for Fig. 7B, the layers not yet PCM-noise-exempted see our standard AIMC crossbar model. While PCM-noise-exempt layers experience no long-term conductance noise, programming errors, or drift, they still are subject to the same cycle-to-cycle read noise, additive output noise, and DAC/ADC quantization choices in our standard AIMC crossbar model.

For ResNet-18, ResNet-50, and DenseNet-121, we find that improving just a few layers can help achieve iso-accuracy ($\mathcal{A}_*^{1h} \geq 99\%$, dashed line in Fig. 7B). This involves only 6.4%, 2%, and 11.3% of the model parameters, respectively (Fig. 7C). Improving MVM fidelity for such a limited number of parameters should prove less costly than across the full DNN. Moreover, we show in Supplementary Notes B.4 that the number of parameters can generally be further reduced—within those most-sensitive layers, only half of the columns need to be PCM-noise-exempted to reach iso-accuracy. However, for the WideResNet-50 DNN, MVM fidelity would need to be further improved, beyond just suppressing PCM weight noise but reducing system noise as well, in order to reach iso-accuracy. Therefore, this particular DNN would require further advances in either HWA training or the overall AIMC specifications, in order to support AIMC deployment without significant accuracy drop. Nevertheless, note that even with the 2% accuracy drop, the WideResNet-50 actually shows the lowest absolute test error among the ImageNet DNNs (see Table 3, e.g., at 1 day), which might make this DNN useful for AIMC deployment despite its significant relative drop (from its own FP baseline).

## Discussion

We have introduced an approach for successfully deploying DNNs onto realistic AIMC inference hardware, at or near iso-accuracy. Our standard AIMC crossbar model incorporates well-known but

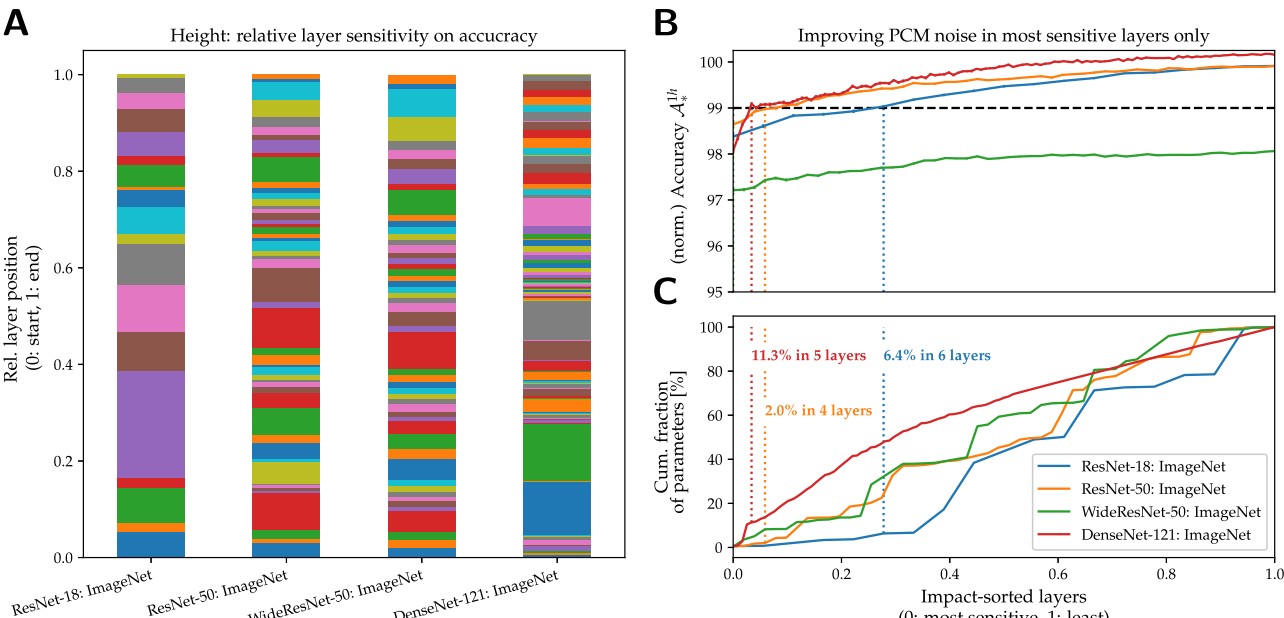

**Fig. 7 | Layer-wise noise impact for ImageNet CNNs. A** Bar-charts reveal the relative impact that different DNN layers have on AIMC accuracy when the PCM conductances in just that layer are made very noisy (overall PCM noise scale set to 15), while all other layers see only minimal PCM noise (overall noise scale set to 0). The height of each bar-segment, arranged in sequential DNN layer order, corresponds to the relative impact of that layer; colors simply delineate layer boundaries. Note that ResNet-50 and WideResNet-50 have very similar graphs since their layers only differ in width. **B** Accuracy $\mathcal{A}_*^{1h}$ as the most critical layers in these four CNNs are exempted from PCM noise, plotted as the fraction of noise-exempt layers is increased, in order from most-sensitive to least-sensitive. **C** Corresponding cumulative fraction of weight-parameters that are PCM-noise-exempted. For ResNet-50, ResNet-18, and DenseNet-121 reducing PCM noise in just a few layers (dotted vertical lines) allows an AIMC crossbar model to achieve iso-accuracy (dashed horizontal line). However, for WideResNet, even non-PCM nonidealities would need to be improved since the residual system noise already causes a significant ($\approx 2\%$) drop in accuracy.

hardware-calibrated nonidealities caused by the analog devices, such as read noise, programming errors, and conductance drift. Going well beyond previous studies, our model also includes nonidealities due to MVM circuit integration, such as system noise, DAC and ADC quantization, and dynamically computed IR-drop. Finally, our model fully addresses the fixed dynamic-range constraints on inputs, weights, and outputs found in all AIMC systems, but previously neglected.

We here investigate the scalability and applicability of the HWA training approach for larger DNNs of various topologies, which mostly have not yet been deployed on actual AIMC hardware due to size constraints of current prototypes. It has been already verified in hardware, however, that HWA training using noise injection is very effective at improving the robustness for selected (smaller) DNNs. For instance in a recent study[54], a ResNet9 CNN was trained with a similar general HWA training approach yielding vastly improved AIMC accuracy in hardware. It remains to be seen whether our simulated iso-accuracy results for the larger-scale DNNs can be verified in hardware in future.

While a few aspects of our study are not directly applicable to hardware designed around non-PCM devices, our standard AIMC crossbar model and our carefully designed inference protocols can readily serve as the basic core for studying such systems (see Supplementary Notes B.2 for a generalization to ReRAM). The intuition we have developed in terms of how various types of noise affect different families of DNN models is also readily transferable.

Some aspects of our AIMC crossbar model have been investigated individually in earlier studies, such as the effect of ADC/DAC quantization, IR-drop, and general read noise[55–57], as well as data-dependent long-term noise[38]. Our main contribution is to combine the long-term data-calibrated noise models of ref. 38 with a more realistic MVM-to-MVM noise model (e.g., quantization, system noise, and IR-drop), and to also include input, weight, and output range restrictions. Moreover, our crossbar model also includes (trainable) digital input and output

scales that, as we show here, improve accuracy of large-scale DNNs when HWA training algorithms are adapted accordingly (see also Supplementary Notes B.3 for an expanded analysis). Since our standard AIMC crossbar model is described here in mathematical detail together with default parameter settings, it should be straightforward to implement it in any modern machine learning or AIMC simulator framework to simulate the expected accuracy upon AIMC deployment. As such, the present work establishes a baseline that can both guide–and be compared against–future AIMC simulation studies. To help make this even more straightforward, our standard AIMC crossbar model has now been incorporated into our open-source AIHWKIT[50,58], which is based on the popular ML framework PyTorch[59], and allows for automatic evaluation of any DNN on AIMC.

However, while our AIMC crossbar model aims at easing the development of new algorithms and their comparisons by establishing a reproducible benchmark, it cannot replace ultimate AIMC hardware verification of the algorithms. Beyond the inevitable variation of design details across different AIMC hardware prototypes, we also use many simplifications and abstractions of the various AIMC nonidealities, since our goal is quick and relatively realistic functional verification of larger DNN workloads. For instance, we assume noise sources are Gaussian, avoiding physically modeled distributions that would be more accurate but significantly slower. We also devised a method to rapidly approximate IR-drop which can adjust dynamically with the input. We chose to intentionally ignore static crossbar effects that would change the conductance value systematically[55,60], since read–write-verify conductance programming can readily adapt to such effects.

Some prior works propose using on-chip or chip-in-the-loop training methods[38,43,49,55,61], which can greatly increase the attainable accuracy by addressing the specific fabrication variations found on that particular chip. However, we strongly believe that the time and cost of such individualized preparation is likely to be untenable for

widespread deployment. Thus in this paper, we have focused on HWA training that can be general enough to be performed once per model per AIMC chip-family, greatly simplifying the deployment onto individual chips. That said, our HWA training approach could readily be combined with more sophisticated online compensation methods, with on-chip or chip-in-the-loop training, or with more than one device pair used per weight, including optimization of how weights are assigned across these conductances[62].

Since HWA training is performed in software before deployment, it has no first-order impact on the latency, throughput or energy efficiency of AIMC hardware. However, as we have shown, HWA training is essential to understanding the tradeoffs between accuracy and these important system performance metrics. For instance, because of the sequential nature of layers of a deep network, shallower but wider layers should generally be preferable for AIMC, since higher utilization of large matrices stored on the crossbar arrays does not significantly change the runtime[52,63] and helps improve energy efficiency. In terms of noise robustness, excessively deep DNNs have disadvantages. Among the ImageNet CNNs tested, DenseNet-121 showed the worst long-term accuracy drop from its FP$_{32}$ model (7.1% in normalized accuracy after 1 year), while WideResNet-50 offered the best raw test error (e.g., 23.76%, versus 24.83% for the next best ResNet-50 at 1 h, see Table 3).

We also find that the RNNs investigated were particularly noise robust. In a complementary recent study[51], a subset of the DNNs investigated here were compared in terms of latency, throughput, and energy efficiency, including the RNN-T, ResNet-50, and BERT-base DNNs. The authors found that the RNN-T was more efficient on a realistic AIMC architecture than the CNNs or transformer models, due to the high utilization as well as reduced need for digital auxiliary operations. Together with our result indicating robustness to nonidealities, RNNs seem highly suited for AIMC. In general, information about performance as well as expected accuracy drop is critical when trying to decide which DNN model to deploy.

A few previous studies have attempted to improve the robustness of DNNs to nonidealities by noise-aware training, where multiplicative or additive Gaussian noise[38,41] is added to weights or activations during training. Similarly, other studies seeking to prevent overfitting or to enhance robustness to adversarial attacks have injected noise into standard floating-point training as a regularization technique[64–70]. While all these methods qualitatively increase the noise robustness of DNNs, the quantitative benefits on real AIMC can neither be accurately reported nor fully optimized by these studies. Since our HWA approach keeps weights mapped in conductance units, a variety of realistic hardware-relevant constraints can be incorporated in a straightforward manner. These include the complexities of PCM programming, and the shallow input-output ranges, IR-drop and quantization affecting the MVM compute—aspects neglected in most previous studies.

We have tried distilling with the FP model as a teacher (similar to ref. 71) and found some benefits when HWA training time is limited. However, since the improvements offered by distilling disappeared at longer training times for most DNN models, we mostly report results without distilling. However, we did find that accuracy with distilling is significantly higher for the Markov model (HMM) Speech LSTM as well as the WideResNet-16 DNN, and these results are shown in Table 3, implying that distilling can be helpful for some DNNs.

Rather than simple Gaussian weight noise[38], we use the expected weight noise distribution characterized from PCM measurements[31] and found it in general superior to other noise structures even when evaluated on an ReRAM-based AIMC evaluation model (see Supplementary Table 5). We find that injection of noise on the weights—together with the correct incorporation of injected longer-term programming noise when modifying the weight matrix during the

backward pass—is crucial for achieving AIMC robustness. One drawback of our approach is that this type of noise injection is currently applied only once per mini-batch, which reduces the effectivity of the noise as batch-size increases. One possible improvement would be to sample the weight noise sources multiple times per mini-batch. Such an extension of our methods should further improve the noise robustness of the HWA-trained DNNs.

In conclusion, we show that comprehensive hardware-aware (HWA) training can greatly enhance the robustness of a variety of deep neural networks (DNNs)—including convolutional neural networks (CNNs), recurrent neural networks (RNNs), and transformers—to the unavoidable device and circuit nonidealities of emerging analog in-memory computing (AIMC) accelerators. In five of the 11 models studied, the techniques we introduce lead to software-equivalent accuracy, defined as 99% of the accuracy-performance offered by the original DNN model beyond random guessing. Averaged across all models, HWA training reduces the gap in model accuracy from 11.3% down to just 1.1% (judged at 1 h).

Through a systematic sensitivity analysis, we identify the nonidealities that are most critical for maintaining accuracy in future system designs. For instance, we observe that nonidealities that effectively add noise to the inputs or outputs—such as ADC and DAC resolution, additive output noise, and S-shaped nonlinearity of the ADC—have the largest impact on DNN accuracy. We also show that certain DNN topologies, such as RNNs, can tolerate more AIMC nonidealities than others. It would be interesting to pinpoint the mechanistic reasons for the increased robustness in particular topologies in future works.

By making this standard AIMC crossbar model available in the open-source AIHWKIT[50], we make it possible for future advances in HWA training techniques to be readily compared to these results. By pinpointing the measures needed to compensate for imperfect AIMC hardware, the tools we have introduced here enable better understanding and optimization of the tradeoffs between model accuracy and desirable performance characteristics such as latency, throughput, and energy efficiency. Continued coordination between HWA training and architectural assessments may even lead to brand-new DNN topologies, specifically designed to maximize the benefits of AIMC hardware—accurate inference at high speed and low power.

## Methods
### AIMC standardized evaluation model
**Affine transform in tile periphery.** We assume that each output column of the analog crossbar has a floating-point scale $\alpha_i$ and offset $\beta_i$ available which implement together an affine transformation. We assume that conductances can be linearly mapped to weight values, so that we can normalize the analog weight values from −1 to 1, corresponding to $-\hat{g}_{max}, \ldots, \hat{g}_{max}$ (see "Weight programming" sub-section). This affine transform then maps the column's physical output (e.g., current), as quantized using an ADC into integers within a certain range, to the value expected by the DNN for the next layer (e.g., activation). Note that such ADC conversion using a scale and bias per column is already available in prototypes[44] but has not previously been incorporated into studies on HWA training.

This digital periphery of an analog MVM can thus be summarized as in Eq. (1), where the operator $\breve{\mathbf{F}} : \mathbb{R}^n \to \mathbb{R}^n$ describes the analog aspects of the AIMC MVM (see (2)), and

$$\text{quant}_b^q(z) \equiv \text{clip}_{-b}^b \left( \frac{2b}{(2^q - 2)} \text{round} \left( \frac{(2^q - 2)z}{2b} \right) \right), \tag{5}$$

describes linear quantization to $2^q - 1$ values in $-b, \ldots, b$ centered around 0. One bin is discarded to force an odd number of bins on either side of zero. Here, $\text{clip}_a^b(x)$ constrains $z$ between minimum $a$ and

maximum $b$,

$$\text{clip}_a^b(z) = \begin{cases} z, & \text{if } a < z < b \\ a, & \text{if } z \leq a \\ b, & \text{if } z \geq b. \end{cases} \tag{6}$$

$\alpha$ is a scalar, per-crossbar value which determines the usable input range. This can either be a learned parameter which is then held fixed during inference (static input range), or can depend dynamically on the current input vector $\mathbf{x}$ (dynamic input range). While main results assume a static input range, we examine performance improvements for the dynamic option (Supplementary Notes B.1).

The scales $\gamma_i$ determine the mapping of conductances to weight values, individually for each crossbar column $i$. During HWA we allow SGD to optimize this parameter, starting from values initialized. $\beta_i$ is used to implement the bias of the MVM, which we implement in digital (FP) precision here. We assume 8-bit quantization, and investigate lower precision as part of our sensitivity analysis.

**Dynamic MVM range.** A critical feature of our crossbar model is that it fully encompasses the finite dynamic-range constraints on inputs, weights and outputs that will be present and unavoidable in any real AIMC implementation. Since both input and weights are normalized within $-1, \ldots 1$ (in analog units), our output-bound setting of $b_{\text{out}} = 10$ means that just 10 fully- on inputs, applied to rows containing maximal-value weights, would fully saturate the output. This is a conservative choice that works for modest-size crossbars and for our assumption that positive current contributions (produced by weight and activation pairs of the same sign) and negative contributions (weights and activations have opposite signs) cancel within the array. This mode is energy-efficient and minimizes IR-drops, but requires the ADC to be capable of measuring bipolar currents[44]. If the crossbar is made much larger, or the positive and negative terms are integrated separately, this may increase energy usage and exacerbate IR-drops, but simplify the ADC design. Furthermore, such choices will likely alter the overall dynamic-range limitations, calling for a reoptimization of $b_{\text{out}}$.

**Analog MVM model.** Our basic model is illustrated in Fig. 3A. The analog MVM $\breve{\mathbf{y}} = \breve{\mathbf{F}}(\breve{\mathbf{x}})$ in Eq. (1) for the quantized, clipped and scaled input vector $\breve{\mathbf{x}} \equiv \text{quant}_1^q(\mathbf{x}/\alpha)$ takes the following general form of Eq. (2), where analog weights $\breve{w}_{ij}(t)$ represent normalized conductances with programming errors, drift, and long-term noise up to time $t_{\text{eval}}$ applied (see "Weight programming"). We include a point-wise non-linear functions $f_i^{\text{NL}}(x)$ to support special cases such as ADC non-linearities; in our standard model, $f_i^{\text{NL}}(x) \equiv x$. Normal random numbers ($\xi_i, \xi_{ij} \sim \mathcal{N}(0,1)$ are drawn for each MVM, representing additive output noise with standard deviation $\sigma^{\text{out}} = 0.04$, and short-term weight noise $\sigma^{\text{W}}(\breve{w})$ that depends on the current weight values (see "Short-term PCM read noise"), respectively. Since the analog output values running from $-10, \ldots 10$ get quantized into digital values from $-127, \ldots 127$ (8-bit), this choice of $\sigma^{\text{out}} = 0.04$ corresponds to almost exactly half of one ADC quantization bin.

**Weight programming.** We adopt a previously described and -characterized weight-programming and drift model for PCM devices[31] as detailed in the following. We assume that the crossbar provides one pair of conductances per weight, where the first (second) member of the device pair is programmed to a conductance between reset (0) and set ($\hat{g}_{\text{max}}$) to handle positive (negative) weights, with the non-active conductance programmed to reset. Only the active conductance is considered in our model. Although recent prototypes support two pairs per weight[44,47], having only one conductance pair increases the

weight density and thus compute efficiency, and poses a more difficult challenge in terms of accuracy and yield.

Each column $\mathbf{w}_i$ of each weight matrix is mapped to a column of target conductances $\hat{\mathbf{g}}_i$. We first initialize each affine scale coefficient using the maximum weight found in that column, $\gamma_i = \max |w_{ij}|$. This allows each weight to be mapped to a scaled target conductance, $\hat{g}_{ij} = \hat{g}_{\text{max}} \frac{w_{ij}}{\gamma_i}$. In our HWA training approach, after this initialization of target conductance and affine scales based on the FP$_{32}$ model weights, we then use SGD to further optimize both the mapped target conductances and scales $\gamma_i$ separately. Table 3 uses this learned weight-to-conductance mapping when evaluating AIMC inference performance.

In a real AIMC system, a positive $\hat{g}$ value gets programmed onto a different physical device than if that particular $\hat{g}$ had been negative. We here assume that only one of the two devices are programmed to particular target conductance whereas the other device is always at reset conductance ($\hat{g}_{ij} = 0$). In this case, one can simplify and compute the MVM directly with signed conductances as done in our model. The programmed conductances $g_{ij}^{\text{P}}$ differ from the desired target values $\hat{g}_{ij}$ as $g_{ij}^{\text{P}} = \hat{g}_{ij} + \sigma^{\text{P}}(\hat{g}_{ij}) \xi$ due to programming noise, assumed to be Gaussian ($\xi \in \mathcal{N}(0,1)$). In turn, the standard deviation of this programming noise depends on the target conductance as

$$\sigma^{\text{P}}(\hat{g}) = c_0 + \sum_{k=1}^n c_k \frac{\hat{g}^k}{\hat{g}_{\text{max}}^k}, \tag{7}$$

where $n = 2$ and $c_0 = 0.26348\,\mu\text{S}$, $c_1 = 1.9650\,\mu\text{S}$, and $c_2 = -1.1731\,\mu\text{S}$, as obtained by fitting to extensive PCM hardware data[31].

**Weight drift and read noise.** Once a PCM device is programmed, the device exhibits both conductance drift and $1/f$ (long-term) read noise. Both are modeled in a statistical manner based on measurements of doped-Ge$_2$Sb$_2$Te$_5$ (d-GST) mushroom PCMs from a large device array integrated in 90 nm CMOS technology[31].

PCM drift: PCM conductance drift, attributed to post-programming structural relaxation, follows an empirical relation

$$g^{\text{D}}(t_{\text{eval}}) = g^{\text{P}} \left( \frac{t_{\text{eval}} + t_0}{t_0} \right)^{-\nu}, \tag{8}$$

where $g^{\text{D}}(t_{\text{eval}})$ is the conductance measured at time $t_{\text{eval}}$ after the programming (assumed to complete at $t_0 = 20\text{s}$[72]) and $\nu$ is the drift coefficient.

The drift coefficients for each device are assumed to be normally distributed, that is $\nu_{ij} \in \mathcal{N}\left(\mu_\nu(\hat{g}_{ij}), \sigma_\nu(\hat{g}_{ij})\right)$, where the mean and standard deviation are empirically determined by fitting to experimental data. The data fits are expressed by a clipped linear function in log-space, that is (with Eq. (6))

$$L(x|a,b,y_{\text{min}},y_{\text{max}}) \equiv \text{clip}_{y_{\text{min}}}^{y_{\text{max}}}(a \ln x + b) \tag{9}$$

where here $x \equiv \frac{\hat{g}}{\hat{g}_{\text{max}}}$. The parameters for $\mu_\nu$ are given by $a = -0.0155$, $b = 0.0244$, $y_{\text{min}} = 0.049$, and $y_{\text{max}} = 0.1$. For $\sigma_\nu$ the parameter are $a = -0.0125$, $b = -0.0059$, $y_{\text{min}} = 0.008$, and $y_{\text{max}} = 0.045$. The drift coefficient $\nu_{ij}$ thus determined for each device are used to model the conductance at any time $t_{\text{eval}}$ using Eq. (8).

PCM read noise: PCM is also known to demonstrate low-frequency noise such as random telegraph noise (RTN) and $1/f^\gamma$ noise with $\gamma \in [0.9, 1.1]$. We follow the empirical noise model of ref. 31, which assumes $\gamma = 1$ and arrives at a read noise standard deviation at time $t_{\text{eval}}$ of ref. 31

$$\sigma_{\text{read}}(t_{\text{eval}}) = \hat{g}\, Q_s(\hat{g}) \sqrt{\ln\left( \frac{t_{\text{eval}} + T_{\text{read}}}{2\, T_{\text{read}}} \right)}, \tag{10}$$

where $Q_s(\hat{g})$ is measured to be

$$Q_s(\hat{g}) = \text{clip}_0^{c_3}\left(c_1\left(\frac{\hat{g}}{\hat{g}_{max}}\right)^{c_2}\right), \tag{11}$$

with $c_1 = 0.0088$, $c_2 = -0.65$, $c_3 = 0.2$.

This read noise is added to the post-drift conductance $g^D(t_{eval})$ to arrive at the final PCM conductance

$$\tilde{g} = \text{clip}_{\hat{g}_{min}}^{\infty}\left(g^D(t_{eval}) + \sigma_{read}(t_{eval})\xi\right) \tag{12}$$

where we set $\hat{g}_{min} = 0$ here and $\xi \sim \mathcal{N}(0,1)$. The weight values $\breve{w}_{ij}$ of the crossbar array for (2) are then obtained by scaling and combining positive and negative parts

$$\breve{w}_{ij} = \frac{\tilde{g}_{ij}}{\hat{g}_{max}}\,\text{sign}\,w_{ij} \tag{13}$$

These long-term PCM effects are applied to all weights prior to the evaluation at time $t_{eval}$ and the weights are subsequently fixed during the evaluation of the test set. Short-term weight noise, redrawn for each MVM, is included separately in Eq. (2) as described in the following paragraph.

Short-term PCM read noise: When evaluating the AIMC DNN at a time $t_{eval}$, the analog weights $\breve{W}$ are established as described in Eq. (13). However, weights are often re-used multiple times during a single input, say across image pixels in a CNN or sequence-tokens in an RNN or transformer model. Here short-term weight noise can cause small but perceptible cycle-to-cycle variations (Fig. 3B).

Modifying the weight matrix at each MVM would be highly inefficient for our HWA training software running on GPUs. To efficiently model such short-term read noise, we use the read noise definition (10) to set $\sigma^w$ in Eq. (2), but refer the resulting noise to the output $\breve{y}_i$. Assuming zero-mean independent normal distributions, we can sum the variances as

$$\tilde{\sigma}_i^W = \sigma_0^W\sqrt{\sum_j |\breve{w}_{ij}|\,|\breve{x}_j|^2}, \tag{14}$$

implying that the weight dependence of the read noise can be approximated as $\propto \sqrt{|\bar{w}|}$. Thus weight noise $\sigma^w$ in Eq. (2) effectively adds $\xi_i\tilde{\sigma}_i^W$ (with $\xi_i \sim \mathcal{N}(0,1)$) to the analog output $\breve{y}_i$. The parameter $\sigma_0^W$ can be identified with $c_1\sqrt{\ln(\frac{\Delta t + t_r}{2t_r})}$ for read noise accumulated over time-period $\Delta t$ (Eq. (10)[31]). Assuming a read duration of $t_r = 250\text{ns}$ and approximate waiting time between two consecutive MVMs ($\Delta t$) to be $100 \times$ longer, we find $\sigma_0^W \approx 0.0175$.

**Drift compensation.** For evaluation times $t_{eval}$ long after NVM programming, the conductance drift Eq. (8) can be compensated in the digital domain without any expensive re-programming[36,73]. This can be done by running a number of analog MVMs on some known test inputs $\{\mathbf{x}^k\}$ immediately after weight programming and recording the overall output magnitude as $s_{ref} = \sum_{ik}|y_i^{(k)}|$. At time $t_{eval}$, just before beginning inference, the same inputs can be applied to measure $s_{eval}$. We then correct the MVM outputs by adjusting the digital $\gamma_i$ (see Eq. (1)) by $\frac{s_{ref}}{s_{eval}}$ to accommodate the average conductance decrease due to drift. We assume one global drift compensation applied to all columns, although this could be done individually at each column if $s_{ref}|_i$ can be measured

sufficiently accurately. Other more sophisticated drift compensation and adaptive refresh methods including in-memory retraining could potentially be applied as well[38].

**Crossbar tile size.** The NVM crossbars available on an AIMC chip are of finite size, typically ranging from $256 \times 256$ (ref. 44) to $512 \times 512$ (ref. 47). We assume a tile size of $512 \times 512$, and assume that enough crossbars are available to support separate crossbars for each weight matrix. Any weight matrix with input dimension >512 is divided into roughly equal parts for programming on as many tiles necessary. Partially used tiles have weights are situated at the bottom of the crossbar, to minimize interference and potential IR-drop, and unused inputs are clamped to zero.

Each tile computes an MVM Eq. (2) using its own periphery Eq. (1). Inter-tile summation is performed at FP precision (FP16), after affine-scaling but before being passed to subsequent digital compute such as activation functions. Because our AIMC nonidealities have no dependencies across output columns, the HWA training code does not need to explicitly break the compute along the output dimension into tile-sized chunks. This helps the simulations run more efficiently on GPUs.

**IR-drop.** Ideally, the voltage along each long bitline in the crossbar would remain constant, so that conductances with the same value could contribute the same current, whether in the farthest or nearest row from where peripheral circuitry is holding the bitline voltage and measuring currents. In a physical crossbar, however, IR-drops imposed by finite wire resistance cause the bitline voltage to vary[74], especially as instantaneous currents get large. To keep the simulation time reasonable, we make a number of approximations when modeling this effect. IR-drop is modeled independently for each crossbar column because any column-to-column differences will be implicitly corrected (to first order) when programming the weight with an appropriate read–write–verify scheme.

However, within each crossbar column, the current contributed by each weight depends on the local bitline voltage, which in turn depends on the other currents being generated elsewhere along the column by that particular input vector. This situation will evolve throughout the integration period due to the pulse-length modulation of those inputs as well as any resulting transients, including the response of the peripheral circuit establishing the bitline voltage. Here, for simplicity and speed of computation for large DNNs, we only consider the average integration current.

The steady-state bitline voltages $\bar{v}_i$ can be computed by solving the equation system

$$(\bar{v}_{i+1} - \bar{v}_i)g_w + g_i^+(v_i^+ - \bar{v}_i) = (\bar{v}_i - \bar{v}_{i-1})g_w + g_i^-(\bar{v}_i - v_i^-) \tag{15}$$

where $g_w$ is the wire conductance between the crosspoint nodes and $g_i^{+/-}$ the weight programmed onto either the positive or negative conductance (with the other programmed into the reset condition, $g = 0$). The individual input voltages, $v_i^-$ and $v_i^+$ of spatially ordered inputs $i$, are linearly prorated from the supply voltages ($v_{ref} \pm V_{read}$) to represent the time-averaged current. The analog output current $\breve{y}$ located at location $i = 0$ is given by $g_w(\bar{v}_0 - v_{ref})$, with $V_{read} = 0.2\,\text{V}$.

This linear system Eq. (15) can be solved by inverting the unique coefficient matrix produced by a given input vector. To speed up the simulation and avoid inverting a $512 \times 512$ matrix for each MVM, we further approximate the solution with a quadratic equation. Thus, in our analog MVM Eq. (2), the IR-drop amount is computed from the

normalized weights and inputs by

$$a_i \equiv \gamma n \sum_j |\breve{w}_{ij}||\breve{x}_j| \tag{16}$$

$$c_i \equiv 0.05\, a_i^3 - 0.2\, a_i^2 + 0.5\, a_i \tag{17}$$

$$\Delta \breve{y}_i^{\text{IR-drop}} \equiv -c_i \sum_j \breve{w}_{ij} \breve{x}_j \left(1 - \left(1 - \frac{j}{n}\right)^2\right), \tag{18}$$

where $\gamma$ is the unitless product of the wire resistance between adjacent cross-points (assumed $0.35\,\Omega$) and the maximal (set) conductance of the device ($g_{\max} = 5\,\mu S$), and $n$ is the number of cross-points occupied by the weight matrix. We assume that smaller weight matrices are located at the lower edge of the crossbar to avoid excess IR-drop. We use Eq. (18) to dynamically approximate the IR-drop across the 512 input channels in Eq. (2) when computing normalized MVM outputs $\tilde{y}$ in all our results. Multiplying these normalized outputs by $g_{\max}V_{\text{read}}$ produces the (time-averaged) physical output currents. To amplify these IR-effects for the sensitivity analysis (Fig. 4), we simply multiply the IR-drop error $\Delta \breve{y}_i^{\text{IR-drop}}$ by a varying scaling factor.

For large inputs where current is flowing throughout the integration window, our estimations using time-averaged current are quite accurate. However, for small inputs where much of the current flow occurs in a small portion of the integration window, instantaneous and average currents differ strongly, and IR-drop will be underestimated. We find that for a Normal distributed weight matrix and random but correlated inputs (as in Fig. 3E), IR-drop deviations are underestimated by roughly a factor of 5. Unfortunately, similar conditions arise across many of our DNNs. Fortunately, our sensitivity analysis (Fig. 4) finds that scaling our time-averaged IR-drop approximation by a factor of >10× does not significantly impact the accuracy of the DNNs, so we can still conclude that DNNs are reasonably robust to IR-drop, albeit by a modest rather than large safety margin. Since IR-drop depends heavily on both on the hardware design (crossbar size, wire resistances, and absolute device conductances) and on the input and weight distributions, detailed circuit-based simulations using the intended workload(s) will remain a critical part of assessing new hardware designs.

### Additional nonlinearities for sensitivity analysis

**PCM device yield.** Emerging memory devices such as PCM exhibit imperfect yield, and some fraction of the devices in a given crossbar array will simply not switch properly[30,75]. PCM devices can end up stuck-at-set ($\hat{g}_{\max}$), stuck-at-reset (conductance set to 0) and stuck-at-random (stuck somewhere between 0 and $\hat{g}_{\max}$). In our sensitivity analysis (Fig. 4), we vary the fraction of failed devices and randomly select their locations.

**S-shaped ADC output nonlinearity.** The output level might gradually saturate more gradually than the desired linear response due to nonlinearity in the ADC[44,76]. To estimate the impact of this for our sensitivity analysis (Fig. 4), we define $f_i^{\text{NL}}$ in Eq. (2) with

$$f_i^{\text{NL}}(z) \equiv \left(1 + \frac{2}{d_{\text{out}}} \sum_{k=1}^{d_{\text{out}}} |\zeta_k|\right)^2 \frac{z}{1 + |\zeta_i z|}, \tag{19}$$

which models a S-shaped saturation with variable slope scaled to approximately cover the full output range. Each of the $d_{\text{out}}$ outputs has an independent ADC and thus a slightly different (pre-determined) shape, $\zeta_i = \mu_\zeta(1 + \sigma_\zeta \xi)$ with $\xi \sim \mathcal{N}(0,1)$ and $\mu_\zeta = \frac{1}{4}$. $\sigma_\zeta$ is only varied in the

sensitivity analysis ("ADC S-shaped nonlinearity"); for our standard AIMC crossbar model, $\mu_\zeta$ and $\sigma_\zeta$ are both set to 0, causing $f_i^{\text{NL}}(z) = z$.

**PCM polarity.** Depending on the hardware and unit-cell design, positive and negative inputs might not create perfectly symmetric read currents. The measured conductance of a PCM device can depend on whether read-current passes from top to bottom electrode, or vice versa. This read-polarity dependence can cause weights to appear systematically altered for negative inputs as compared to positive inputs. Although the average effect can be corrected by adjusting read voltages, device-to-device or conductance-dependent variations can remain. To model this effect in our sensitivity analysis, we separate positive and negative inputs into two phases (setting a negative input to 0 in the positive phase and vice versa), and scale each weight in the negative phase by $(1 + a_{ij})$ where $a_{ij} \sim \mathcal{N}(0, \sigma_a)$. We then vary this nonideality parameter $\sigma_a$ as "weight asymmetry std."

### MVM error calculation

To quantify the fidelity of the analog MVM, we calculate the expected deviation of the analog MVM as compared to the ideal MVM as MVM error $\epsilon_M$, defined by the relative normalized deviations (see ref. 77)

$$\epsilon_M(W, \{\mathbf{x}_k\}) = \frac{\langle ||\mathbf{y}_k - \tilde{\mathbf{y}}_k||_2 \rangle_k}{\langle ||\mathbf{y}_k||_2 \rangle_k}, \tag{20}$$

where $\mathbf{y}_k = W\mathbf{x}_k$ is the ideal MVM output to input vector $\mathbf{x}_k$ using matrix $W$, and $\tilde{\mathbf{y}}$ is the actual AIMC output considering all hardware-related nonidealities as defined in Eq. (1).

The MVM error is obviously zero if the AIMC is equal to the ideal outcome, but otherwise it depends on both the particular weight matrix $W$ and set of input vectors $\mathbf{x}_k$ used to estimate Eq. (20). To best reflect the impact of the nonidealities on the DNN, inputs $\mathbf{x}_k$ should ideally be taken from the distribution of actual input activation vectors, and $W$ should be the target weight matrix, for the specific DNN layer in question.

However, to quantify the MVM error independent of the DNN in question, we calculate the standard MVM error $\epsilon_M^*$ by using normal distributed weights, $w_{ij} \sim \mathcal{N}(0, 0.246)$ and uniform inputs $x_i \sim \mathcal{U}(-1, 1)$ with a tile size of $512 \times 512$. For our standard AIMC crossbar model as described in "AIMC standardized evaluation model", the standard MVM error is $\epsilon_M^* = 15\%$ (not considering drift).

### AIMC hardware-aware DNN training

Robustness to the nonidealities of AIMC inference hardware can be improved by hardware-aware (HWA) training—a DNN retraining method that applies expected nonidealities to the forward pass of the SGD, with the backward pass performed using regular FP precision.

Our HWA training approach is to use the general form of the expected analog MVM nonidealities as described in Eq. (2), together with the injection of the expected programming errors (but without any conductance drift). Further, we use the HWA training step to also establish the digital peripheral parameters of Eq. (1), in particular the static input range $\alpha$ (see "Learning the input range") and the weight-to-conductance mapping $\gamma_i$ (see "Learning of weight-to-conductance conversion factors"). In addition, we find that ramping up the injected programming error strength (see "Re-training with weight noise injection"), fixed scales and individual learning rates per tile (see "Learning of weight-to-conductance conversion factors"), weight-clipping (see "Weight mapping and clipping") and distilling (see "Distilling with floating-point teacher") improved the robustness and achievable accuracy in the presence of AIMC nonidealities.

In general, the HWA training starts from an already FP-trained DNN, and hyper-parameters (learning rate, injected noise strength) are

optimized. We verified the effectiveness of our HWA training approach on the very same DNNs used in a previous study[38] and found, on average, a >10% decrease in AIMC test error for long $t_{eval}$ times. This directly indicates the improvement of our approach over previous methods (see Table 2).

In the following paragraphs, our HWA training methods are presented in more detail.

**Retraining with weight noise injection.** Injecting noise to improve robustness to nonidealities was suggested by a number of studies[38,40,41], and has been one of the hallmarks of HWA training for AIMC. In previous studies, noise has been injected in multiple ways, such as output[38,40], input[38], or weight noise[38,41]. Different types of weight noise distributions have been used, such as additive (scaled by the current maximal weight[38]) or multiplicative[41] Gaussian.

Methods for injecting weight noise have differed across previous studies. For instance, Joshi et al.[38] added newly drawn Gaussian weight noise to the weight matrix reversibly for each image input (not mini-batch) only during the forward pass (and not during backward pass which was done with the actual weight matrix). However, it is more mathematically correct to also apply these same weight perturbations during the backward pass (but not to the reference weights to which updates are applied), as is commonly done for weight regularization techniques such as drop-connect[78]. Furthermore, although the exact noise injection method (input, output, or weight noise) does not seem to matter much[38], generic additive Gaussian noise does not conform with the expected AIMC noise structure. For instance, PCM programming errors are actually conductance-value dependent and not just additive.

Here, we improve on the earlier approaches in the following ways: First, rather than just a generic noise term, we apply all expected non-idealities and hardware design choices (given by Eq. (2)) into the HWA retraining. This includes dynamic-range limitations, system noise, and analog-digital conversions—all previously ignored. We inject weight noise in a mathematically consistent way to both forward and backward passes, redrawing from random distributions once per mini-batch. We draw the weight noise from the (scaled) expected programming error distribution including 20 s of PCM read noise (see Eq. (7) and Eq. (10), respectively) instead of using generic additive or multiplicative Gaussian distributions. We find that injecting PCM noise structure improves the HWA training across DNNs in comparison to other noise injection strategies, even when testing for other memory technologies (see also Supplementary Notes B.2 for an in-depth analysis). Finally, the scale of the injected weight noise is a hyper-parameter and ramped up linearly over a number of epochs, which we found to improve the HWA training. See Supplementary Methods A.1 for the detailed hyper-parameters and noise settings used for each DNN.

**Learning of weight-to-conductance conversion factors.** To achieve a good weight-to-conductance conversion, we train the $\gamma_i$ scale factors in Eq. (1) using SGD. To improve the HWA training, it is beneficial in most DNNs to represent these scale factors by $\gamma_i = \tilde{\gamma}_i \, \kappa$, where both the column-wise $\tilde{\gamma}_i$ and per-tile $\kappa$ factors can be learned. We treat the learning of either factor as a hyper-parameter for a particular DNN. In case of not learning, $\gamma_i$ is initialized by the weight mapping (see "Weight mapping and clipping") and $\kappa$ is set to 1.

In case of CNNs, where the matrix-sizes vary widely, the learned values $\tilde{\gamma}_i$ are uniquely scaled for each weight matrix by a fixed $c_{aws}$ value, which re-scales the learning rates per tile so that the trained parameters can all have similar magnitude $\approx 1$. This auto-weight scaling factor, $c_{aws}$, is set to the value suggested by the Xavier weight initialization[79,80], $c_{aws} = \sqrt{\frac{3}{n}}$, where $n$ is the input dimension of the weight matrix.

If $\kappa$ is learned, we encourage the learning of larger outputs and weights by down-scaling the output range to $[-1, 1]$ which typically improves the signal-to-noise ratio, thus $\kappa = \frac{\tilde{\kappa}}{b_{out}}$. Here $b_{out}$ is the fixed output bound of Eq. (1), and $\tilde{\kappa}$ is a per-tile learnable scalar which is initialized to $b_{out}$ (and is subject to weight decay).

Note that during inference evaluation, the digital periphery can simply apply one scale factor per output column, since the various scale factors described above can be re-combined after the completion of HWA training.

**Weight mapping and clipping.** Since we use the output scales $\gamma_i$ to keep the analog weights $\breve{w}_{ij}$ of Eq. (2) mapped in (normalized) conductance units (within $-1, ..., 1$), the FP weights $w_{ij}$ of the trained DNN need to be mapped to conductances before initiating HWA training. For that, we set initially

$$\breve{w}_{ij} \leftarrow \frac{w_{ij}}{\max_j |w_{ij}|} \tag{21}$$

$$\gamma_i \leftarrow \max_j |w_{ij}| \tag{22}$$

so that $\gamma_i \breve{w}_{ij} = w_{ij}$.

We keep training from creating excessively large analog weights. $\breve{w}$, by clipping after each update to this same range. In some cases (see Supplementary Methods A.1), we encourage learning of larger analog weights to maintain signal-to-noise ratio by remapping weights according to Eq. (21) once every epoch.

**Learning the input range.** The input range clipping bound $c_{input}$ in Eq. (1) is learned during HWA training. To encourage a smaller clipping value (and thus a more compact input distribution), a decay is introduced. To augment the gradient update for the clipping bound, we scale gradient updates by the current bound value. For small datasets (such as for transformer fine-tuning tasks), the HWA training is too short to learn the clipping bound value from scratch. In such cases, we initialize $c_{input}$ to the average absolute maximal value of the input vectors over a number of mini-batches before starting HWA training, subject to a cap (nominally $\max(c_{input}) = 10$).

**Distilling with floating-point teacher.** If the model output dimension is large, such as for the LSTM models with large vocabulary size, the HWA training greatly benefits from distilling with the FP model. In knowledge distillation[81], an already trained "teacher" model augments the usual one-hot labels with expected class probabilities, which can drive a "student" model to a good solution more rapidly than when training only with the one-hot label vectors. We use the distilling applied at the last layer, with the FP model without any AIMC non-idealities as the teacher and the HWA training as the student. The temperature controlling the distribution of pseudo-probabilities was fixed to 10, and training loss was weighted by a mixture of 75% from the distillation and 25% from the regular loss.

## HWA training experiments

We applied and optimized the HWA training process described in this section to a variety of AI workloads—including text prediction, speech-to-text translation, and image classification—as listed in Table 1. In general, our HWA training approach addressed these DNNs similarly, since a too DNN-specific retraining approach would be impractical. In Supplementary Methods A.1, we detail any specific differences used in the HWA training of these DNNs, including learning rates and injected noise strength. We select the last available rather than the best checkpoint, and we repeat experiments multiple times and average the results to obtain repeatable results.

## Data availability

The training and test datasets used for this study are publicly available[82–87]. The raw data that support the findings of this study can be made available by the corresponding author upon request after IBM management approval.

## Code availability

The full simulation code used for this study cannot be publicly released without IBM management approval and is restricted for export by the US Export Administration Regulations under Export Control Classification Number 3A001.a.9. However, the open-source Apache License 2.0 IBM Analog Hardware Acceleration Toolkit (AIHWKit) at https://github.com/IBM/aihwkit[88] implements and reproduces the full AIMC inference model evaluation using the same simulation engine. The HWA training simulations can be reproduced using the AIHWKIT in a very similar manner as described here.

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

## Acknowledgements

We thank the IBM Research AI Hardware Center and Rensselaer Polytechnic Institute for access to the AIMOS supercomputer, and the IBM Cognitive Compute Cluster for additional compute resources. We would like to thank Syed Ghazi Sarwat for help with the bipolar AIMC model, and Timothy Phillips, Julian Büchel, Corey L. Lammie, Fabio Carta, Kaoutar El Maghraoui, Irem Boybat-Kara, Stefano Ambrogio, Tayfun Gokmen, and Omobayode Fagbohungbe for fruitful discussions.

## Author contributions

M.J.R., M.L.G., C.M., H.T., A.S., and V.N. conceived the study; M.J.R., M.L.G., C.M., H.T., A.C., S.R.N., O.F., N.L., and A.S. contributed to the development of the hardware-aware training approach; A.C., M.J.R., F.O., and H.T. conducted the HWA training simulations for transformers, M.J.R. for ImageNet DNNs, C.M. and M.J.R. for LSTM, M.L.G. and M.J.R. for HMM-LSTM, M.J.R. and S.R.N. for CIFAR CNNs, and A.F. and M.J.R. for RNN-T networks; P.N. and G.W.B. contributed to the IR-drop model and interpretation; M.J.R. and C.M. conducted to the sensitive analysis, the impact of weight distribution analysis, and the CNN layer analysis; M.J.R. conducted the ReRAM simulations and all supplemental analyses; M.J.R. developed the simulator software framework; M.J.R., G.W.B., M.L.G., C.M., A.S., A.C., A.F., and H.T. contributed to writing and editing of the manuscript.

## Competing interests

The authors declare no competing interests.
