## [Peer Review File · Nature Communications]

REVIEWER COMMENTS

Reviewer #1 (Remarks to the Author):

The authors present an analog in-memory computing model for simulating phase-change memory crossbar-based neural network implementations with comprehensive modeling of nonidealities related to quantization, IR-drop, device variation, and other factors. This model is used for hardware-aware training, which achieves state-of-the-art robustness in training DNNs for analog implementation. Furthermore, this model is used to analyze robustness of 11 different DNN topologies to nonidealities with the authors' hardware-aware training technique. The analysis of DNN topology robustness and the effects of hardware-aware training on achieving iso-accuracy is exhaustive and useful for determining best applications of analog in-memory computing technology. The simulation tools are made available in AIHWKit, which will provide other researchers opportunities for more accurate simulations for informing high-accuracy and efficient analog hardware. The models are described in detail in the methodology section and are technically accurate. Beyond the analysis provided in the paper, some power analysis would be a helpful addition, exploring for example the effects of hardware-aware training or certain nonidealities on power consumption. Demonstrating some of the same analysis on another memory technology would also support the claim that the model presented can "readily be extended to other types of NVM devices". One potential weakness of the paper is the clarity of some sections. Figure 2 is not well explained and can be confusing to the reader. Similarly, Figure 3 shows much data that doesn't always seem to be referred to in the text. Some of the data (particularly the blue data points) aren't explained at all. The equations (such as Eq. 6) that are referred to early on in the text aren't explained until the methodology section, which can make the paper difficult to follow on a first read-through. Clarifying some of these figures and giving a preview of the equations early on in the text can significantly help the paper's clarity. The paper also has a few very minor typos throughout that should be proofread.

Reviewer #2 (Remarks to the Author):

This work presented a hardware-aware-training (HAT) framework with a standard MVM behavior model for analog in-memory computing (AIMC) architectures and accomplished versatile experiments on a variety of AI workloads systematically. Meanwhile, this work presented comprehensive analysis of different nonidealities of AIMC systems on different AI models. But the methods discussed in this work are very mature techniques – HAT and Improving AIMC fidelity of selected layers, which limit its novelty. And there are all simulation results and lacked real chip's results to prove the improvement of proposed HAT. What's more, many technical details about MVM operations using PCM devices are discussed in this work. Above all, I think this work is not

suitable for Nature Communications, but might consider a technical journal. Besides, some concerns should be addressed, which are listed as following:

1. The proposed standard model contains many specific characteristics about PCM, such as conductance drift, and weight programming, which are different from other AIMC devices, such as RRAM, FeFET, so how to apply the standard model to other AIMC devices?
2. There are many proposed simulation tools for AIMC, such as NeuroSim, which also contains many circuits and devices non-ideal factors, what are the differences between this work's standard model with those proposed models? There need some detailed comparisons compared to prior works.
3. There are many non-ideal factors in the proposed standard model. Are all factors calibrated with hardware results respectively or only is the final model formula (Eqs.6 in Methods) calibrated with hardware results?
4. The scaling factors (α , γ , β) are all optimized during training process, which might be floating point number. So how to consider the circuits overhead when keeping these parameters during inferencing? And most of current AIMC accelerators do not contain the floating point units (FPUs), so what degree of recovered accuracy will be achieved using this proposed HAT method when inferencing without the FPUs?
5. There is a mistake in the sentence - "Figure 3B shows that, even after including PCM drift, the effective MVM error of the standard AIMC crossbar-model we will use throughout the paper roughly corresponds to 4bit fixed-point quantization of weights or inputs" in page 4. The description of this sentence seems to correspond to Figure 3D.

Reviewer #3 (Remarks to the Author):

This paper presents a simulation-based exploratory analysis of the accuracy of AIMC (analog in-memory computing) for common AI workloads across 11 different DNN topologies. The paper investigates the sensitivity and robustness of AIMC to unavoidable nonidealities and introduces a new realistic crossbar model based on previously published results from fabricated PCM devices. This significantly improves earlier retraining approaches presented in the literature.

The paper performs a detailed analysis of various nonidealities and demonstrates that some previously understudied device and peripheral circuit nonidealities, as well as input and output noise, are more detrimental to DNN accuracies than usually-considered device variations in crossbar arrays. These are new significant results of the paper, which make it highly valuable to the wide deep learning hardware acceleration research community.

The paper concludes that various topologies of DNNs, including CNNs, RNNs, and transformers, can be successfully retrained on AIMC to reach accuracies on par with or fairly close to floating-point trained systems. It shows that RNNs are particularly robust to nonidealities that add noise to the inputs or outputs of crossbar arrays. These are noteworthy conclusions substantiated with new interesting, and comprehensive results.

Strengths:

The authors pose several important questions on the way of answering the larger more general question 'Would AIMC work on real-world large deep learning models?' They provide simulation results of these in their methodology and include wait times to showcase the impact of conductance drift in their utilised PCM-based crossbars.

Within Fig. 3 the effect of various nonidealities using the modelling of physical PCM devices is shown. Next, in Table 2, the authors show how these nonidealities degrade the performance of various DNNs even when the digital parameters of the crossbars are adjusted. This substantiates the need for a method to account for these nonidealities.

The main contribution of the paper is considering and consolidating a wide realistic array of nonidealities, compared to the established literature that mainly covers a limited subset of nonidealities and does not include detailed input/output noise analysis. The authors provide their open-source analog hardware kit platform, with a set of parameters that can account for these different variations, which are realistically set based on physical PCM-based crossbars, while also accounting for circuits such as DAC and ADCs.

One of the other contributions of this work is the proposed weight mapping to crossbar arrays, which helps reduce the MVM error due to PCM device nonidealities and consequently improves the DNN error.

Limitations:

This study is all based on software simulations in the analog hardware kit platform by IBM. It would be very interesting to see how the proposed simulated HWA training actually works and compliments the training of a fully hardware-based small CNN, such as the ResNet32, which due to its lower number of parameters may be possibly mapped and implemented on the team's PCM crossbars. The IBM team seem to be one of the few groups who have the logistics and capabilities to achieve such a challenging target, so this reviewer hopes that in this work, or in a future study this happens.

Section 7 proposes to improve overall AIMC-based DNN accuracy, by reducing noise on selected more sensitive layers in more challenging architectures such as wide and deep CNNs. The authors show that in some instances only 2% noise-free parameters can improve the error to within 1% of FP DNNs. However, it is not clear how this 2% of parameters are spread in the most sensitive layers. Are these all the parameters in these layers, or are some of the parameters within these layers?

In addition, the authors do not comment on the cost of having these noise-free PCMs, when techniques such as averaging tile results from multiple passes or having multiple conductance pairs are used.

One of the discussion points of the article is investigating and understanding trade-offs between model accuracy, latency, throughput, and energy efficiency. However, such investigations are not present in the paper, and given the software nature of the article, these can be done in the revised version.

Another interesting discussion point that could be added is around the reasoning for the higher susceptibility of CNNs compared to RNNs in the investigated architectures.

Minor comments:

References should be numbered in the order they appear in the text.

The full stop at the end of the caption of Fig. 1 is on the next line.

The statement “any nonlinearities within the analog circuitry performing summation will further degrade MVM precision” is not accurate, because the nonideality or nonlinearity in analog circuits

such as ADCs “can” sometimes change the random MVM result for the better. Therefore, the sentence is better changed to “... can further degrade ..”

Fig. 3(D). Did you really want to show the model output after 1-second conductance drift or the blue dashed line is showing it after 1h drift? It seems to be the latter because 1s is not significant.

In the sentence “Figure 3B shows that, even after including PCM drift, the effective MVM error of the standard AIMC crossbar model we will use throughout the paper roughly corresponds to 4-bit fixed-point quantization of weights or inputs” it seems that you mean Fig. 3D and not 3B. Please check.

Batch sizes for the experiments in Table 2 are not provided.

In Section A.1 you write “.... range α (see Sec. A.1).” This should be “see Fig. 2.”

REVIEWER COMMENTS

Reviewer #1 (Remarks to the Author):

The authors present an analog in-memory computing model for simulating phase-change memory crossbar-based neural network implementations with comprehensive modeling of nonidealities related to quantization, IR-drop, device variation, and other factors. This model is used for hardware-aware training, which achieves state-of-the-art robustness in training DNNs for analog implementation. Furthermore, this model is used to analyze robustness of 11 different DNN topologies to nonidealities with the authors' hardware-aware training technique. The analysis of DNN topology robustness and the effects of hardware-aware training on achieving iso-accuracy is exhaustive and useful for determining best applications of analog in-memory computing technology. The simulation tools are made available in AIHWKit, which will provide other researchers opportunities for more accurate simulations for informing high-accuracy and efficient analog hardware. The models are described in detail in the methodology section and are technically accurate.

We thank the reviewer for the accurate summary of our manuscript.

Beyond the analysis provided in the paper, some power analysis would be a helpful addition, exploring for example the effects of hardware-aware training or certain nonidealities on power consumption.

We agree with the reviewer that power analysis of certain nonidealities would be very interesting. Since HWA training is done prior to AIMC deployment, it does not have a *direct* impact on the performance. However, for some selected nonidealities – such as the possibility to reduce ADC energy requirements because a better HWA-trained model could potentially maintain accuracy despite reduced ADC precision – energy efficiency can certainly be impacted *implicitly* during AIMC deployment.

In general, however, a careful analysis of the performance would require a specific definition of the underlying hardware architecture, which would go far beyond this current study. Moreover, in the current manuscript we intentionally chose to focus on the robustness of different medium-to large-scale DNNs to AIMC noise, and on improving pre-training algorithms to increase this robustness, because we felt these topics have been somewhat understudied in the AIMC field. We decided this was best done while remaining agnostic to the specific underlying hardware architecture.

We also note that a subset of our authors on this manuscript have recently published a study on the latency, energy efficiency, and throughput of a particular hardware architecture assumption for a subset of the DNNs studied here. We thus refer both the reviewers and our readers to that complementary paper (Jain et al. TVLSI 2023) for a more detailed discussion on performance. For your convenience, we have reproduced their Fig. 6 as Fig. 1 in this response document. In the manuscript, we also expanded the discussion related to performance, writing:

Figure 1: Figure 6 from Jain et al. 2023 TVLSI. (a) Energy efficiency (green bars) of across some of the same DNNs studied in the manuscript in respect to accuracy. Throughput depends on batch size because of auxiliary digital operations. (b) Latency for batch size 1.

“We also find that the RNNs investigated were particularly noise robust. In an complementary recent study [Jain et al 2023 TVLS], a subset of the DNNs investigated here were compared in terms of latency, throughput, and energy-efficiency, this kind of information on available accuracy gain including the RNN-T, ResNet-50, and BERT-base DNNs. The authors found that the RNN-T is more efficient on a realistic AIMC architecture than the CNN or Transformer models, due to the high utilization as well as reduced need for digital auxiliary operations. Together with our result indicating robustness to nonidealities, RNNs seem highly suited for AIMC. In general, information about performance as well as expected accuracy drop is critical when trying to decide which DNN model to deploy.”

Demonstrating some of the same analysis on another memory technology would also support the claim that the model presented can “readily be extended to other types of NVM devices.”

Many thanks for this good suggestion. We have now added an extensive analysis using an AIMC inference model based on ReRAM instead of PCM to the appendix (see Section ‘Generalizing to other memory technologies’). For these studies, we used the same chip-design assumptions as in the main text, however, we replaced all PCM-related parameters (weight noise and drift) with a custom weight-noise model fitted to published ReRAM data from Wan et al. (Nature 2022). Interestingly, we found that DNNs which were robust to PCM were generally also robust to ReRAM, suggesting that our pre-training methods generalize to other memory technologies as well. This also suggests that any future studies which evaluate, extend and optimize training algorithms on the suggested PCM-based AIMC crossbar model should readily benefit other memory technologies as well. In more quantitative detail, the accuracy drop when using ReRAM rather than PCM was slightly more pronounced for some DNNs, likely because of the higher measured noise within the intermediate conductance range, as reported by the available published ReRAM data. It would be interesting to study such effects – hopefully incorporating more extensive ReRAM characterization data – as well as potential algorithmic remedies, in more detail in the future.

One potential weakness of the paper is the clarity of some sections. Figure 2 is not well explained and can be confusing to the reader. Similarly, Figure 3 shows much data that doesn’t always seem to be referred to in the text. Some of the data (particularly the blue data points) aren’t explained at all. The equations (such as Eq. 6) that are referred to early on in the text aren’t explained until the methodology section, which can make the paper difficult to follow on a first read-through. Clarifying some of these figures and giving a preview of the equations early on in the text can significantly help the paper’s clarity. The paper also has a few very minor typos throughout that should be proofread.

Indeed, the choice to defer almost all equations to the Methods might not have been ideal. In the revision, we have now followed the reviewer’s suggestion and expanded the Section ‘Analog IMC standard MVM model’ in the Results to include the main equations (former Eq.3 and Eq.6) in the main text and leave only the more detailed discussion and derivation to the Methods.

Moreover, we added a paragraph describing Fig. 2 in more detail and updated the figure itself, which is an illustration of the abstract AIMC crossbar model that we define by the equations. We have added:

“Thus, as illustrated in Fig. 2, digital FP inputs x_i are scaled by a scalar α , quantized in a fixed range (by the DAC), and then subject to the non-ideal analog computation with noisy weights constrained by a fixed weight range (grey bell curves), as well as an additive system noise (blue bell curves). The (noisy) outputs of the analog crossbar \hat{y}_i are then digitized again by parallel ADC in a fixed output range, and finally re-scaled and shifted by the combined digital FP scales $\gamma_i\alpha$, and offsets β_i , respectively.”

Reviewer #2 (Remarks to the Author):

This work presented a hardware-aware-training (HAT) framework with a standard MVM behavior model for analog in-memory computing (AIMC) architectures and accomplished versatile experiments on a variety of AI workloads systematically. Meanwhile, this work presented comprehensive analysis of different nonidealities of AIMC systems on different AI models.

Many thanks for the compact summary.

But the methods discussed in this work are very mature techniques – HAT and Improving AIMC fidelity of selected layers, which limit its novelty.

We agree with the reviewer that hardware-aware (HWA) training *per se* was done previously, as we acknowledge by citing the previous papers. As we clarify in the introduction, these previous papers only showed HWA training for very selected small set of small-size DNNs and did not address the scalability of the technique to larger DNNs and DNNs of various topologies. Since all earlier methods introduced their own incompatible hardware definitions, the accuracy drop obtained in individual DNNs is impossible to compare between these various studies. Here – for the first time to our knowledge – we look at a large selection of larger-scale DNNs of various topologies and use the same AIMC model for assessing the accuracy drop in a quantitative and reproducible manner. We feel this is essential for fair evaluation of any HWA algorithmic approaches, and for informative testing of AIMC suitability of different DNN topologies. Furthermore, we also add adequate AIMC realism to the evaluation for large-scale DNNs in simulations such as system noise, IR drop and dynamic range limitations — largely neglected in earlier simulations studies.

In another main contribution, we provide a significant number of novel algorithmic improvements over previous HWA training approaches, such as training the input ranges as well as trainable column-wise output scalings, as well as more realistic noises during training. We also describe how the conductance value distributions change, providing critical intuition as to why and how HWA-trained models can offer improved robustness.

Finally, the investigations of the relative impact of different non-idealities on the accuracy on a large number of DNNs has not been attempted before either. Our analysis finds that input and output related non-idealities have a larger impact on accuracy in all topologies. We also show the individual limits of AIMC non-idealities on various DNNs in a comparable manner, offering insight and guidance for future chip designs. This part of our study can serve as a starting point and baseline reference for development of new algorithms targeting improved robustness in specific areas of AIMC.

We thus think that our study indeed provides a large number of novel techniques as well as important insights into the scalability and improvement of hardware-aware training across DNN topologies.

In summary, given the importance of HWA training for AIMC deployment, this manuscript doesn't just advance the capabilities of HWA training and quantify the robustness of a diverse set of large-scale DNNs. It also establishes the foundation for a standardized approach for AIMC evaluation, suitable for benchmarking of future HWA training algorithm developments. This has not been feasible before, because previous studies have relied on very disparate hardware assumptions that have either been too simple to be useful, or too dependent on proprietary hardware inaccessible to other researchers.

In contrast, our AIMC model is easy enough to be straight-forwardly implemented in standard ML framework, and efficient enough to use on large-scale DNNs, but also realistic enough that it can provide predictive and relevant results. Ease of benchmarking is at the core of the recent explosion of new approaches in the wider field of deep learning and we feel that a similar benchmarking approach is very important for algorithmic advancement in the field of AIMC as well. With a rigorous quantitative reference using a repeatable evaluation model, algorithmic advancement becomes readily measurable — allowing rapid progress to be made by a community of researchers working collectively.

And there are all simulation results and lacked real chip's results to prove the improvement of proposed HAT.

As detailed also in the previous response, one of our main aims, next to the improvement of HWA training in general, is to provide a baseline reference for future algorithmic improvements. While adding specific hardware results would indeed show what is possible for that particular chip and device technology in question, such hardware verification would not serve the purpose of building a repeatable reference model for algorithmic studies. Others will not be able to use the same proprietary hardware and reproduce the results. In fact, even if we showed improvement of one particular hardware instance, it can still be unclear how well such improvements will generalize to other hardware designs.

Furthermore, our paper is forward-looking, investigating a large number of large and diverse DNNs that exceed the capacity of available chip prototypes and thus are difficult if not impossible to evaluate in hardware presently. We believe it is crucially important to provide a widely reproducible baseline for judging future HWA training improvements that is independent from actual hardware instance, allowing noise-sources and idealities to be individually amplified or suppressed to an extent incompatible with any hardware-based demo.

Nevertheless, we calibrated our AIMC model using data of detailed device measurements, which were published earlier, and thus expect the accuracy in simulation to strongly correlate with our standard model. However, the quantitative accuracy number will always vary slightly for individual chip design choices and memory technologies. For emerging memory technologies still under active research and development, results can even vary from wafer to wafer, or from chip to chip on the same wafer. This means that having a highly repeatable software model becomes even more important for independent algorithmic advancement.

What's more, many technical details about MVM operations using PCM devices are discussed in this work. Above all, I think this work is not suitable for Nature Communications, but might consider a technical journal. Besides, some concerns should be addressed, which are listed as following:

1. The proposed standard model contains many specific characteristics about PCM, such as conductance drift, and weight programming, which are different from other AIMC devices, such as RRAM, FeFET, so how to apply the standard model to other AIMC devices?

We thank the reviewer for the very good suggestion to show how our standard AIMC model could be made more applicable to other memory technologies as well. Indeed, some aspects of the model are tailored to PCM devices, namely the programming error, weight noise, and drift. To show how the model can straightforwardly be adapted to other memory technologies, we use ReRAM statistics from a recent paper Wan et al. (Nature 2022), and replace the PCM-based parts of the model by a fit to that data. This shows that our results generalize well to other analog memory devices. These new results have been added in a new section (Supplementary Information B.2). More specifically, we show how the HWA-trained DNNs are affected by this change (see Supplementary Tab. 4) and also investigate how adapting the weight noise distribution during HWA training changes the results (Supplementary Fig.1 and Supplementary Tab.5).

2. There are many proposed simulation tools for AIMC, such as NeuroSim, which also contains many circuits and devices non-ideal factors, what are the differences between this work's standard model with those proposed models? There need some detailed comparisons compared to prior works.

Indeed, there are a number of simulation frameworks which aim to simulate the non-idealities in AIMC. Our model mainly features:

1. Conductance-dependent short-term and long-term noise and drift models calibrated on PCM data
2. Implementation of (trainable) digital input and output scaling-factors that vastly improve the AIMC accuracy for large-scale models

3. Realistic input / output / weight range restrictions
4. ADC/DAC quantization, cycle-to-cycle noise, and IR-drop

Some aspects are also included in previous approaches (e.g. RxNN Jian et al. 2020; NeuroSim Peng et al. 2021). For instance, NeuroSim also features ADC/DAC, IR-drop as well as (uniform) read noise. However, that tool does not include the digital scales necessary for adequate conductance mapping for large-scale DNNs, and also does not have data-calibrated, conductance-dependent weight noise and drift models for inference. Moreover, it is in general more focused on performance / latency estimation rather on (inference) accuracy and algorithmic improvements, and also typically assumes that weights are sliced into sub-bits, which would require more area as in our model where weights are represented by analog conductance values instead. We have not seen any usage of NeuroSim or other CIM frameworks for transformers and RNNs (or any reference in their documentation), which implies these increasingly important DNN topologies are not yet supported. Note that adding the AIMC MVM simulation correctly into the RNN loop is not straightforward and requires implementing a custom LSTM / RNN layer in modern ML frameworks.

However, our main goal in this article when defining a standard AIMC model is not to promote a particular simulator, but rather to give a detailed, reasonable, and realistic setting for simulating AIMC, that is fully characterized with parameter settings and equations in the article and thus reproducible. Our AIMC standard model is thus independent of any given CIM simulator framework and could be readily implemented with any ML framework with the information provided in the article.

To emphasis this point and our contributions better, we have now added the following sentence in the discussion:

“Some aspects of our AIMC crossbar-model have been investigated individually in earlier studies, such as the effect of ADC/DAC quantization, IR-drop, and general read noise [Jain et al. 2021, Peng et al. 2019, Xia et al. 2017], as well as data-dependent long-term noise [Joshi et al. 2020]. Our main contribution is to combine the long-term data-calibrated noise models of [Joshi et al. 2020] with a more realistic MVM-to-MVM noise model (e.g. quantization, system noise, and IR-drop), and to also include input, weight, and output range restrictions. Moreover, our cross-bar model also includes novel (trainable) digital input and output scales that, as we show here, greatly improve accuracy of large-scale DNN when HWA training algorithms are adapted accordingly (see also Supplementary Notes B.3 for an expanded analysis). Since our standard AIMC cross-bar model is described here in mathematical detail together with default parameter settings, it should be straightforward to implement it in any modern machine learning or CIM simulator framework to simulate the expected accuracy upon AIMC deployment. [...]”

3. *There are many non-ideal factors in the proposed standard model. Are all factors calibrated with hardware results respectively or only is the final model formula (Eqs.6 in Methods) calibrated with hardware results?*

The weight noise related to PCM is calibrated from measured PCM devices in an earlier publications as cited [Nandakumar et al. 2019]. Other non-idealities, such as the output noise and the dynamic range and ADC / DAC resolutions are based on common hardware assumptions. For instance, 8 bit ADC / DAC is in line with common chip prototypes [Khaddam-Aljameh et al. 2021] and the output noise is chosen on the level of the bin width of the ADC. The dynamic-range ratio of input-range times weight-range divided by output range is set to be 10, which is relatively shallow and chosen based on the requirements of the DNNs, where we show with our presented results that this range is enough, given the other constraints. Since some of these assumptions might vary from hardware-to-hardware and other design choices, we extensively explore the effect

of changing these non-idealities on the accuracy of the DNNs (see Fig. 5 and Fig.6). We are trying to bridge the gap to be generic enough for different designs but also realistic enough for evaluating AIMC. As also discussed in an earlier response, we feel it is important to establish rather generic and reproducible standard models for AIMC in particular for the ML community to help benchmarking the algorithmic developments, which is complementary to studies that focus on precise modelling of a specific hardware instance.

To clarify the AIMC model parameter choices, we now write in the Results Section:

“We mainly investigate the situation where all weight related parameters have been carefully calibrated to existing PCM hardware [Nandakumar et al. 2019], however, the model can be adapted to other memory technologies as well (see Supplementary Notes B.2). Quantization levels (8-bit, e.g. [Khaddam-Aljameh et al. 2021]) and system noise (on the order of the ADC bin-width) are set to reasonable values by default, however, we will also explore their impact in a sensitivity analysis.”

4. *The scaling factors (α, γ, β) are all optimized during training process, which might be floating point number. So how to consider the circuits overhead when keeping these parameters during inferencing? And most of current AIMC accelerators do not contain the floating point units (FPUs), so what degree of recovered accuracy will be achieved using this proposed HAT method when inferencing without the FPUs?*

We thank the reviewer for this valuable insight. We here investigate large and diverse DNN topologies. These DNNs have a variety of layers that cannot be straightforwardly mapped onto AIMC chips that do not feature any FPUs. For instance, Transformers have matrix-matrix multiplications of two activations that are assumed to happen in a FPU, as AIMC requires (weight) stationarity to play out its strength. Layer-norms (that are part of Transformers) need a standard-deviation computation even during inference. Some DNNs (LSTMs, Transformers) have non-linear Sigmoid, Tangent, or GeLU activations, that cannot easily be done in integer numbers. It is nearly impossible to support such diverse set of DNNs without having any flexible FPU cores as part of the chip. Indeed, larger AIMC chips are designed using multiple FPUs and other cores next to AIMC crossbars to handle these auxiliary operations (see e.g. Jain et al. 2023 TVLSI).

Thus, at least a scalar number conversion to FP after each AIMC MVM needs to be supported. We indeed assume a column-wise conversion, that is multiple γ_i that could be replaced in principle by a single factor that would reduce the memory requirements. However, these factors can additionally be used to implement other required factors e.g. that are part of a subsequent batch normalization layer, as well as help to calibrate the ADCs and thus serve multiple purposes. We have now added an in-depth investigation in the Supplementary Information (Section B.3 ‘Tile-wise digital output scale instead of column-wise digital output scales’) about whether these factors could in principle be compressed into integer numbers plus a single FP scale (without re-training; Supplementary Fig. 2) or replaced by a single factor. We show that having only a single factor would instead need a modified HWA training that would increase the HWA training complexity and cost (Supplementary Tab.7). Our analysis also shows that incorporating these factors γ_i (or any batch norm factors) into the conductances (after HWA training) results in a much worse robustness in most (large) models and is thus not advisable (Supplementary Figure 2), in particular when FP-trained DNNs are directly mapped (Supplementary Tab.6) .

The α serves the input range mapping into the fixed range DAC and is required for number conversion. If only ReLU activation functions were to exist between AIMC layers, the digital number output could indeed be kept in integer, and α potentially not needed. However, to keep our AIMC model general and flexible enough, we assume FP inputs and outputs throughout, as we expect to have non-linear FP computation between layers in general.

We assume that the β_i will be used to represent the bias of the MVM, and thus are needed for most AIMC layers. In principle, bias could be added to the AIMC crossbar and represented in conductances, however, unless multiple rows of conductances are used, the available conductance

range would be too limited and DNNs are typically very sensitive to errors in the bias. Moreover, FPU typically can handle a scalar multiplication plus bias simultaneously and thus having digital FP biases will not add any more runtime penalty. Finally, these biases can again be used for ADC calibration, which typically have slightly different offsets after fabrication.

Note that performance, latency, and throughput using these design assumption have been investigated in detail in a recent study (Jain et al. 2023 TVLSI).

5. There is a mistake in the sentence - “Figure 3B shows that, even after including PCM drift, the effective MVM error of the standard AIMC crossbar-model we will use throughout the paper roughly corresponds to 4bit fixed-point quantization of weights or inputs” in page 4. The description of this sentence seems to correspond to Figure 3D.

Many thanks for noticing this typo. It is now corrected.

Reviewer #3 (Remarks to the Author):

This paper presents a simulation-based exploratory analysis of the accuracy of AIMC (analog in-memory computing) for common AI workloads across 11 different DNN topologies. The paper investigates the sensitivity and robustness of AIMC to unavoidable nonidealities and introduces a new realistic crossbar model based on previously published results from fabricated PCM devices. This significantly improves earlier retraining approaches presented in the literature.

The paper performs a detailed analysis of various nonidealities and demonstrates that some previously understudied device and peripheral circuit nonidealities, as well as input and output noise, are more detrimental to DNN accuracies than usually-considered device variations in crossbar arrays. These are new significant results of the paper, which make it highly valuable to the wide deep learning hardware acceleration research community.

The paper concludes that various topologies of DNNs, including CNNs, RNNs, and transformers, can be successfully retrained on AIMC to reach accuracies on par with or fairly close to floating-point trained systems. It shows that RNNs are particularly robust to nonidealities that add noise to the inputs or outputs of crossbar arrays. These are noteworthy conclusions substantiated with new interesting, and comprehensive results.

Strengths:

The authors pose several important questions on the way of answering the larger more general question ‘Would AIMC work on real-world large deep learning models?’ They provide simulation results of these in their methodology and include wait times to showcase the impact of conductance drift in their utilised PCM-based crossbars.

Within Fig. 3 the effect of various nonidealities using the modelling of physical PCM devices is shown. Next, in Table 2, the authors show how these nonidealities degrade the performance of various DNNs even when the digital parameters of the crossbars are adjusted. This substantiates the need for a method to account for these nonidealities.

The main contribution of the paper is considering and consolidating a wide realistic array of nonidealities, compared to the established literature that mainly covers a limited subset of nonidealities and does not include detailed input/output noise analysis. The authors provide their open-source analog hardware kit platform, with a set of parameters that can account for these different variations, which are realistically set based on physical PCM-based crossbars, while also accounting for circuits such as DAC and ADCs.

One of the other contributions of this work is the proposed weight mapping to crossbar arrays, which helps reduce the MVM error due to PCM device nonidealities and consequently improves the DNN error.

We thank the reviewer for the accurate summary and overall positive evaluation of our contributions.

Limitations:

This study is all based on software simulations in the analog hardware kit platform by IBM. It would be very interesting to see how the proposed simulated HWA training actually works and compliments the training of a fully hardware-based small CNN, such as the ResNet32, which due to its lower number of parameters

may be possibly mapped and implemented on the team’s PCM crossbars. The IBM team seem to be one of the few groups who have the logistics and capabilities to achieve such a challenging target, so this reviewer hopes that in this work, or in a future study this happens.

We agree with the reviewer that ultimately the accuracy obtained needs to be verified on hardware. However, we argue here that this hardware verification should be seen as a separate and parallel effort to algorithmic advances.

One of our goals in this manuscript is to provide a reproducible reference and guidance to algorithmic advances for a wider (ML) community that do not have access to immediate hardware. The idea is that multiple hardware-aware training approaches can be quickly tested and ranked against our proposed “standard” AIMC inference evaluation even without access to hardware. It indeed needs to be shown that DNNs that are trained robustly against our AIMC crossbar model, are also ranked similarly on a particular AIMC hardware, however, there are many slightly different hardware prototypes and while the general ranking should be similar, the actual quantitative accuracy drops will always differ slightly between chip architectures and memory technologies.

We have now added an additional analysis that uses a modified AIMC model based on ReRAM and we find that there is a high correlation of the robustness against testing on the AIMC-based crossbar model (see Supplementary Notes B.2). We thus think that optimizing and benchmarking against our standard AIMC model will benefit in robust algorithms and DNNs for all AIMC technologies.

Besides, given the size, complexity and broad range of the diversity of the investigated DNNs, we think that hardware verification of their accuracy is well beyond the scope of the current study that focuses solely on the algorithmic part of hardware-aware training. However, in future works, we certainly aim to measure the accuracy of similar DNNs on a hardware prototype.

Section 7 proposes to improve overall AIMC-based DNN accuracy, by reducing noise on selected more sensitive layers in more challenging architectures such as wide and deep CNNs. The authors show that in some instances only 2% noise-free parameters can improve the error to within 1% of FP DNNs. However, it is not clear how this 2% of parameters are spread in the most sensitive layers. Are these all the parameters in these layers, or are some of the parameters within these layers?

We thank the reviewer for this interesting suggestion. In the analysis of Figure 7, all parameters in the selected, most sensitive layers are PCM-noise-exempted. However, we agree with the reviewer that in principle a subset of the parameters within individual layers might be more relevant than the others. To investigate this possibility, we have now added an additional analysis in Supplementary Notes B.4, where we look at individual columns to be PCM-noise-exempt instead of full layers. Exempting only some columns of a layer would indeed further decrease the additional area requirements for e.g. repeating these columns to improve their noise properties. As shown in Supplementary Tab. 8, we find that when ordering the columns with their average absolute gradient magnitude and selecting a subset of columns with maximal values, only about half of the columns are needed to be PCM-noise-exempt instead of all to reach iso-accuracy (in ResNet-18 and ResNet-50; for DenseNet-121 somewhat more than half of the columns are needed to reach iso-accuracy, see Supplementary Tab. 8 for more details). Thus, one can generally further reduce the number of PCM-noise-exempt parameters with this sub-layer approach.

We added the following sentence into the main text:

“Moreover, we show in the Supplementary Notes B.4 that the number of parameters can generally be further reduced — within those most sensitive layers, only half of the columns need to be PCM-noise-exempted to reach iso-accuracy.”

In addition, the authors do not comment on the cost of having these noise-free PCMs, when techniques such as averaging tile results from multiple passes or having multiple conductance pairs are used.

Using multiple conductance pairs (here we assume only one pair) will increase the chip area and ultimately reduce the overall storage capacity. However, if e.g. only some of the layers (or even selected columns) are needed to be less noisy, improving the noise spec might actually come almost “for free” for instance if some weight matrices do not utilize all conductance values of a crossbar of fixed size. For instance for convolutions, kernel matrices are often elongated with more inputs than outputs, e.g. 128×512 . In that case, columns could be multiplied 4 times on the same crossbar without (significant) run-time impact (summing outputs need to be added in digital). 4 times repetition would already reduce the NVM noise by about $\sqrt{4} = 2\times$.

Therefore, while completely noise-free serves more as an upper bound analysis, reducing the noisiness of some selected layers is actually relatively cheap, with very little run time impact (as long as under-utilized crossbars are available).

Alternatively, some of the most sensitive layers may also simply be computed on FPUs in FP precision. Note that we have already made the (common) assumption that the first and the last layer is performed in FP precision (in line with most studies for quantization aware training for reduced-precision digital AI accelerators). However, how costly this would be in terms of energy and throughput, would need to be investigated with many chip architecture assumptions (similar to Jain et al TLSVI 2023), which would go beyond this study.

One of the discussion points of the article is investigating and understanding trade-offs between model accuracy, latency, throughput, and energy efficiency. However, such investigations are not present in the paper, and given the software nature of the article, these can be done in the revised version.

A detailed discussion of energy efficiency, throughput and latency is tied to a particular chip architecture that turns out to be very complex for these DNNs as they feature a complex mixture of analog and digital operations. A detailed performance analysis would therefore be beyond the scope of the study that focuses on the algorithms and noise robustness.

However, our manuscript is actually complementary to a recently published study (with involvement of some of the authors from this manuscript) that tries to estimate latency, energy efficiency, and throughput in detail for a particular AIMC chip architecture and features some of the very same DNNs we looked at in terms of accuracy drop (ResNet-18, ResNet-50, BERT-base, RNN-T; see Jain et al. 2023 TVLSI). For your convenience, we have reproduced Figure 6 from that paper in Fig. 1 in this document. In that study, it becomes clear that the energy efficiency as well as throughput and latency for RNNs is superior to CNNs, while transformers are somewhat in between (improving with larger batch sizes). Together with the findings of our study that RNNs are actually very noise robust, it suggests that RNNs seem to be ideally suited for AIMC architectures.

We have added a number of sentences to the discussion to point out this fact and reference better to the complementary TVLSI study:

“We also find that the RNNs investigated were particularly noise robust. In an complementary recent study [Jain et al 2023 TVLS], a subset of the DNNs investigated here were compared in terms of latency, throughput, and energy-efficiency, including the RNN-T, ResNet-50, and BERT-base DNNs. The authors found that the RNN-T is more efficient on a realistic AIMC architecture than the CNN or Transformer models, due to the high utilization as well as reduced need for digital auxiliary operations. Together with our result indicating robustness to nonidealities, RNN seem highly suited for AIMC. In general, information about performance as well as expected accuracy drop is critical when trying to decide which DNN model to deploy.”

Another interesting discussion point that could be added is around the reasoning for the higher susceptibility of CNNs compared to RNNs in the investigated architectures.

We agree that this would be an interesting topic. At this point, we can only speculate as it might have multiple reasons. For instance, the data sets might be simply easier to interpret than other

tasks. Another idea could be that the bounded activation functions (e.g. Tangent and Sigmoid) actually are more noise robust, since they always operate in the same range and squash outliers. However, showing any of these ideas convincingly would require a more in-depth analysis and we thus refrain from raising this topic in the discussion and leave it for future studies.

To highlight this interesting future topic we added the following sentence to the conclusion:

“It would be interesting to pinpoint the mechanistic reasons for the increased robustness in particular topologies in future works.”

Minor comments:

References should be numbered in the order they appear in the text.

We thank the reviewer for noticing the wrong order. We have now corrected the formatting in the revision to better reflect the Nat. Comm. requirements including the citation order, bold figure and table caption titles, and the correct format of references to the Supplementary Information.

The full stop at the end of the caption of Fig. 1 is on the next line.

Corrected.

The statement “any nonlinearities within the analog circuitry performing summation will further degrade MVM precision” is not accurate, because the nonideality or nonlinearity in analog circuits such as ADCs “can” sometimes change the random MVM result for the better. Therefore, the sentence is better changed to “... can further degrade .. ”

We agree with the reviewer that in exceptional cases nonidealities might actually be beneficial.

We have changed the wording as suggested.

Fig. 3(D). Did you really want to show the model output after 1-second conductance drift or the blue dashed line is showing it after 1h drift? It seems to be the latter because 1s is not significant.

We thank the reviewer for noticing this inaccuracy. Indeed, it should be 1h in (D) corresponding to the 1h points in (A). We have changed the figure accordingly.

In the sentence “Figure 3B shows that, even after including PCM drift, the effective MVM error of the standard AIMC crossbar model we will use throughout the paper roughly corresponds to 4-bit fixed-point quantization of weights or inputs” it seems that you mean Fig. 3D and not 3B. Please check.

Indeed. We have corrected this typo in the revision.

Batch sizes for the experiments in Table 2 are not provided.

Batch sizes were indeed somewhat hidden in the Supplementary Methods. We have now add a table for the hyper-parameters to make these more visible. See Supplementary Table 1.

In Section A.1 you write “... range α (see Sec. A.1).” This should be “see Fig. 2.”

Indeed. We have corrected it accordingly.

REVIEWER COMMENTS

Reviewer #1 (Remarks to the Author):

My previous comments have been adequately addressed and I believe this article is suitable for publication.

Reviewer #2 (Remarks to the Author):

The author replied most of my concerns in detail and corrected the mistakes. This manuscript aims to provide a hardware-independent AIMC model for algorithm designers, and the authors suggest that it is a fair way to judge the improvement on future AIMC-oriented algorithm designs. The insight of this manuscript is very interesting and forward-looking. But there is one more thing which is still unclear. The HAT method was proposed actually to reduce the accuracy loss from hardware non-ideal factors when inferencing. If there was no hardware result to prove this AIMC model's capability, why can get a convinced training results from this AIMC model for an algorithm designer, who may be not familiar with AIMC characteristics? What's more, the authors indeed did versatile experiments on simulations of a variety of large main-stream neural networks, but it was not very straightly to show this AIMC model's capability. As author's reply shown, it is not practical to implement a very large model on current in-memory computing systems, but some small and typical models could be demonstrated, which could give some more intuitive results to the readers. Even though current AIMC system were varied rapidly, and the test results on a real system could be varied from chip to chip, the real system's results are also meaningful for users when using the proposed AIMC models. Because the way to get the AIMC model parameters on different AIMC systems can be generalized. The test results are on your real systems and correspond to one group of specific AIMC model parameters for your systems. Considering a very realistic scenario, if you describe the detailed method to get the AIMC model parameters on your one specific hardware, it can be reproducible for others to get their AIMC parameters and train their models for their hardware, which could also improve their inferencing results on chip.

Reviewer #3 (Remarks to the Author):

In their revision, the authors have done a good job addressing most of this reviewer's questions and concerns. Although they have not included all the suggestions made by this reviewer, their argument

on this exclusion is reasonable in that their work is mainly aimed at providing a reproducible reference for algorithm development for the community with no access to hardware platforms. They have also mentioned, as I hoped, that in their future work, they will look at how their proposed simulated HWA training actually works and compliments the training of a fully hardware-based DNN.

The authors have also added new information on similar papers that address some of my questions and have included new supplementary and experimental data to address my other questions.

Overall, I see this paper as a valuable software-based architectural simulation effort, which would be very beneficial to the overall ML hardware and memristive computing communities.

REVIEWER COMMENTS

Reviewer #1 (Remarks to the Author):

My previous comments have been adequately addressed and I believe this article is suitable for publication.

We appreciate the positive evaluation of our revised manuscript.

Reviewer #2 (Remarks to the Author):

The author replied most of my concerns in detail and corrected the mistakes. This manuscript aims to provide a hardware-independent AIMC model for algorithm designers, and the authors suggests that it is a fair way to judge the improvement on future AIMC-oriented algorithm designs. The insight of this manuscript is very interesting and forward-looking.

We thank the reviewer for the positive evaluation of our revision.

But there is one more thing which is still unclear. The HAT method was proposed actually to reduce the accuracy loss from hardware non-ideal factors when inferencing. If there was no hardware result to prove this AIMC model's capability, why can get a convinced training results from this AIMC model for an algorithm designer, who may be not familiar with AIMC characteristics?

We agree that achieving good accuracy using actual hardware is certainly the ultimate goal. Here we took the approach to first clearly define a hardware AIMC model (that is based on previously published AIMC hardware data), and show the impact of the AIMC nonidealities on large-scale models as well as develop a HWA training method showing how to increase robustness.

Thus, for any AIMC hardware that approximates our AIMC model, our simulations show and quantify the impact different noise sources have on accuracy. Furthermore, any hardware can be matched against our assumptions (i.e comparing the MAC error (Eq. 20), the level of output noise, weight noise etc.) and therefore one can use our nonideality variations to estimate the expected accuracy on a given hardware instance. Thus, our results can guide future hardware developments without actually performing large-scale hardware experiments.

Of course, our AIMC model does make a number of simplifications and might not match any given hardware perfectly, and thus hardware experiments are still extremely important. We however think that these hardware experiments are a separate effort reserved for future publications. Successively, our model can be improved and extended to include even more nonidealities, once they are measured in different hardware realizations.

What's more, the authors indeed did versatile experiments on simulations of a variety of large main-stream neural networks, but it was not very straightly to show this AIMC model's capability. As author's reply shown, it is not practical to implement a very large model on current in-memory computing systems, but some small and typical models could be demonstrated, which could give some more intuitive results to the readers.

It is already well established and verified for selected DNNs that similar (but much simpler) HWA-training methods (i.e. weight noise injections) result in better accuracy in hardware (see e.g. [Joshi et al., Nat.Comm., 2020] and [Khaddam-Aljameh et al., JSSC, 2022] that both show experimental hardware inference results using weights trained with noise injection). A more recent study by Le Gallo et al. (<https://arxiv.org/abs/2212.02872>, accepted in Nature Electronics) demonstrates experimental hardware inference results on a ResNet-9 network having almost 2M weights which was trained with the IBM AIHWKIT using the general HWA approach presented in this paper (with some adapted settings). The HWA trained network achieved 92.81% accuracy on hardware, < 1% below the software baseline of 93.67%. In comparison, the same network without HWA training (simply using floating-point pre-trained weights) achieves 75.16% accuracy, which is approximately 19% lower than the baseline.

In fact, in our manuscript we use the same small ResNet32 DNN for verification of our new methods, which has been already employed on hardware (see [Joshi et al., Nat.Comm., 2020]), and show that our HWA-training methods improves the results (see Supplementary Table 2). While it would indeed be interesting to establish that this improvement in simulation translates to improvement in that particular hardware instance, we think that it would not yield significantly more value to our current manuscript as our hardware model assumptions are also slightly different from the earlier paper (more forward-looking design assumptions).

More importantly, while we also improve the HWA-training methods here to be able to handle flexibly various DNN topologies, in this manuscript we are mainly focusing on generalizing the HWA training approach to larger scale DNNs and measuring the relative impact of AIMC nonidealities of different AI workloads (e.g. Transformer, LSTMs, CNNs). As we pointed out, implementing even one of the large-scale DNNs in actual hardware is a very difficult achievement in itself, and this would go well beyond the current scope of this study.

To make these points clearer and also add the recent citation above, we have added the following paragraph to the discussion:

We here investigate the scalability and applicability of the HWA-training approach for larger DNNs of various topologies, which mostly have not yet been deployed on actual AIMC hardware due to size constraints of current prototypes. It has been already verified in hardware, however, that HWA-training using noise injection is very effective of improving the robustness for selected (smaller) DNNs. For instance in a recent study [Le Gallo et al. 2022], a ResNet9 CNN was trained with a similar general HWA-training approach yielding vastly improved AIMC accuracy in hardware. It remains to be seen whether our simulated iso-accuracy results for the larger scale DNNs can be verified in hardware in future.

Even though current AIMC system were varied rapidly, and the test results on a real system could be varied from chip to chip, the real system's results are also meaningful for users when using the proposed AIMC models. Because the way to get the AIMC model parameters on different AIMC systems can be generalized. The test results are on your real systems and correspond to one group of specific AIMC model parameters for your systems.

We certainly agree with the reviewer that hardware results are a very important final benchmark, we simply see our manuscript complementary to that effort. To emphasize this important point, we have adapted and extended the following paragraph in the discussion. It now reads:

However, while our AIMC crossbar-model aims at easing the development of new algorithms and their comparisons by establishing a reproducible benchmark, it cannot replace ultimate AIMC hardware verification of the algorithms. Beyond the inevitable variation of design details among different AIMC hardware prototypes, we also use many simplifications and abstractions of the various AIMC nonidealities, since our goal is quick and relatively realistic functional verification of larger DNN workloads.

Considering a very realistic scenario, if you describe the detailed method to get the AIMC model parameters on your one specific hardware, it can be reproducible for others to get their AIMC parameters and train their models for their hardware, which could also improve their inferencing results on chip.

We agree with the reviewer that showing how to get the best and most robust DNN for a particular hardware instance is a worthwhile endeavour to achieve the best possible accuracy. However, note that training of individual DNNs for a particular AIMC hardware instance would be too costly in practice as it is not scalable.

We alluded to this aspect in the Discussion, where we write:

[..], we strongly believe that the time and cost of such individualized preparation is likely to be untenable for widespread deployment. Thus in this paper, we have focused on HWA training that can be general enough to be performed once per model per AIMC chip-family, greatly simplifying the deployment onto individual chips.

Reviewer #3 (Remarks to the Author):

In their revision, the authors have done a good job addressing most of this reviewer's questions and concerns. Although they have not included all the suggestions made by this reviewer, their argument on this exclusion is reasonable in that their work is mainly aimed at providing a reproducible reference for algorithm development for the community with no access to hardware platforms. They have also mentioned, as I hoped, that in their future work, they will look at how their proposed simulated HWA training actually works and compliments the training of a fully hardware-based DNN.

The authors have also added new information on similar papers that address some of my questions and have included new supplementary and experimental data to address my other questions.

Overall, I see this paper as a valuable software-based architectural simulation effort, which would be very beneficial to the overall ML hardware and memristive computing communities.

Reviewer 3 Associate Prof Mostafa Rahimi Azghadi, PhD

Many thanks for the positive evaluation of our revised manuscript.

REVIEWERS' COMMENTS

Reviewer #2 (Remarks to the Author):

My previous comments have been addressed well. No further comment. I recommend the acceptance of this paper in Nature Communications.

REVIEWER COMMENTS

Reviewer #2 (Remarks to the Author):

My previous comments have been addressed well. No further comment. I recommend the acceptance of this paper in Nature Communications.

We thank the reviewer for the positive assessment of our revision and for the time and effort for the thorough review.